# Architecture Matters: Uncovering Implicit Mechanisms in Graph Contrastive Learning

**Xiaojun Guo**[1]*    **Yifei Wang**[2]*    **Zeming Wei**[2]    **Yisen Wang**[1, 3]†

[1]National Key Lab of General Artificial Intelligence,
School of Intelligence Science and Technology, Peking University
[2]School of Mathematical Sciences, Peking University
[3]Institute for Artificial Intelligence, Peking University

## Abstract

With the prosperity of contrastive learning for visual representation learning (VCL), it is also adapted to the graph domain and yields promising performance. However, through a systematic study of various graph contrastive learning (GCL) methods, we observe that some common phenomena among existing GCL methods that are quite different from the original VCL methods, including 1) positive samples are not a must for GCL; 2) negative samples are not necessary for graph classification, neither for node classification when adopting specific normalization modules; 3) data augmentations have much less influence on GCL, as simple domain-agnostic augmentations (e.g., Gaussian noise) can also attain fairly good performance. By uncovering how the implicit inductive bias of GNNs works in contrastive learning, we theoretically provide insights into the above intriguing properties of GCL. Rather than directly porting existing VCL methods to GCL, we advocate for more attention toward the unique architecture of graph learning and consider its implicit influence when designing GCL methods. Code is available at `https://github.com/PKU-ML/ArchitectureMattersGCL`.

## 1 Introduction

Over the past few years, Self-Supervised Learning (SSL) has emerged as a promising approach towards utilizing abundant real-world data without costly human-annotated information [34, 9, 54, 35]. Among various SSL techniques, contrastive learning (CL) has established itself as the avant-garde framework for self-supervised visual representation learning [5]. By pulling similar samples near and pushing dissimilar samples far apart, contrastive learning is able to learn semantically expressive representations and achieves huge success on multiple downstream tasks [5, 17, 4].

Inspired by the success of contrastive learning in the visual domain, contrastive learning is also extensively explored on graph data and achieves competitive performance to supervised ones [40, 26, 65]. Despite technical varieties, most graph contrastive learning (GCL) methods share the same high-level skeleton as visual contrastive learning (VCL). Generally speaking, GCL applies augmentations to generate different views of the original graph, and learn node or graph representations by contrasting positive and negative data pairs. Though some works argue that GCL requires domain-specific designs for graph [29, 59, 10], the design of GCL generally obeys the same paradigm as VCL with three key components: data augmentations, positive pairs for feature alignment, and negative pairs for feature uniformity.

---

*Equal Contribution.
†Corresponding Author: Yisen Wang (yisen.wang@pku.edu.cn)

37th Conference on Neural Information Processing Systems (NeurIPS 2023).

In this paper, we challenge the commonly held beliefs regarding GCL by revealing its distinct characteristics in comparison to VCL. Specifically, we perform a systematic study with a wide range of representative GCL methods on well-known benchmarks and find three intriguing properties: 1) for the positive pair part, we demonstrate that GCL can achieve competitive performance without adopting any positive pairs. This finding stands in stark contrast to VCL, which dramatically fails in the absence of positive samples. 2) for the negative pair part, we observe that for the graph classification task, GCL performs well without any special designs in the no-negative setting, which is a notable departure from VCL. 3) for the data augmentation part, for VCL, elaborate data augmentations are indispensable in boosting performance [5, 55]. However, we find that GCL is relatively robust under vanilla domain-agnostic data augmentations (*e.g.,* random Gaussian noise).

To explain these intriguing properties of GCL, we delve deep into the model architecture, and uncover the interesting interplay between model components and graph contrastive objectives. **First**, we shed light on the implicit regularization mechanism of graph convolution in GCL. As is known, graph convolution encourages neighbor nodes to have similar features during propagation. We rigorously show that graph convolution implicitly minimizes a neighbor-induced alignment loss, which reveals its complementary relationship with the learning objective and elucidates the positive-free GCL. **Second**, we highlight the role of the projection head in GCL without negative samples or any specific designs, where we find for graph classification, the projection head implicitly selects a low-rank feature subspace to satisfy the loss. **Third**, for the node classification, by incorporating a normalization layer capable of driving nearby node features apart, we show that, without uniformity loss, a GCN encoder alone can prevent features from collapsing to a single point. We theoretically characterize this by connecting this normalization layer with a neighbor-induced uniformity loss.

These intriguing distinctive properties of GCL reveal that the design of contrastive learning can be very domain-specific. Importantly, as also shown in previous works [48, 37], the contrastive algorithm has an implicit interaction with the architecture. Therefore, experiences obtained from one domain (*e.g.,* images) should not be directly considered universal. Instead, when designing graph self-supervised learning methods, we should consider the unique properties of graphs and graph-based models. We hope that our findings could inspire a better understanding of graph contrastive learning and pave the way for new approaches to self-supervised learning on graphs.

We summarize our main contributions below:

- We perform comprehensive evaluations of popular GCL methods across various benchmarks, and find intriguing and general properties of GCL compared with VCL: 1) GCL works without positive samples; 2) GCL works without negative samples on graph classification task; 3) GCL shows less dependence on delicately designed augmentations, where random Gaussian noise even works.

- We reveal the reasons behind the intriguing properties of GCL by theoretically uncovering the interplay between contrastive learning objectives and model architectures: 1) We shed light on the implicit regularization mechanism of graph convolution by establishing its connection with a neighbor-induced alignment objective; 2) We show the collapse-preventing effect of ContraNorm in the no-negative setting by theoretically characterizing its connection with a neighbor-induced uniformity loss.

- Our findings appeal to a re-examination of the real effectiveness of each component in GCL, and provide new insights for designing graph-specific self-supervised learning.

## 2 Related Work

**Contrastive Learning.** Contrastive learning has attracted intensive attention in self-supervised learning. Its primary objective is to learn a space where similar pairs are closely clustered while dissimilar pairs are far apart. The introduction of Information Noise Contrastive Estimation (InfoNCE) [32] propels contrastive learning to a new climax in the vision domain. Methods such as SimCLR [5] have achieved performance comparable to supervised methods on vision tasks by leveraging stronger augmentations, larger batch size, and a non-linear projection head. MoCo [17] employs a dynamic queue and a moving-averaged encoder to generate more negative samples. The heavy computational burden imposed by a large number of negative pairs prompts the search for alternatives in contrastive learning. BYOL [11] addresses this by enforcing diversity in representations through an asymmetric

architecture and stop-gradient techniques. Barlow-Twins [69] proposes a feature regularization method that maximizes agreements between different views of a sample while eliminating redundancy. On the understanding of contrastive learning, Wang and Isola [53] highlight alignment and uniformity as key properties of contrastive loss, where alignment refers to the closeness of positive pairs and uniformity pertains to the uniform distribution of normalized features on the hypersphere. Wang et al. [56] establish the relationship between contrastive learning and graph neural networks. Zhuo et al. [80] analyze one kind of contrastive learning without negative samples like BYOL from the perspective of rank difference. Zhang et al. [72] reveal the connection between contrastive learning and masked image modeling. There are also other works studying how extra information helps contrastive learning [7, 73].

**Graph Contrastive Learning.** Stimulated by the success of contrastive learning on images, assorted contrastive attempts are applied to graphs. Motivated by Deep InfoMax (DIM) [19], DGI [51] learns by maximizing mutual information between node representations and corresponding high-level summaries of graphs. InfoGraph [42] adopts the DIM principle on the graph classification task. Inspired by SimCLR, GRACE [77] maximizes the agreement of corresponding node representations in two augmented views for a graph. Similarly, GraphCL [66] learns graph-level representations by maximizing the global representations of two views for a graph. Building upon these pioneer efforts, a plethora of GCL methods have been proposed recently [59, 30, 10, 45, 79]. Inspired by the sparsest cut problem, SCE [74] introduces a Laplacian smoothing trick and then proposes a model without positive samples but only using negative samples for training. It is noted that SCE adopts a specific design of the backbone network and learning objectives (details in Appendix A.2). Different from the above methods, our work does not focus on designing a specific graph contrastive method but generally discusses whether components like positive samples and negative samples are needed in the context of graph contrastive learning.

To relieve the learning from negative samples, some methods are also been proposed. Borrowing the idea of BYOL, BGRL [46] scales up graph contrastive learning to large-scale datasets. Along another line, Bielak et al. [2], Zhang et al. [71] add feature regularization objectives to make the cross-correlation matrix between two views close to an identity matrix, ensuring the two representations are distinguishable. However, these works are all proposed as a specific design that can not be straightforwardly transferred to other GCL methods.

In GCL, augmentations have also emerged as a subject of intensive research. Basic augmentations depend on the node features and topology information, *e.g.,* node feature masking, edge perturbation, and subgraph sampling. Existing methods mostly adopt an empirical combination of basic graph augmentations [77, 66], which is found to benefit more than augmentation of a single type [66]. Adaptive selections of augmentations with learnable strategies have also been proposed [43, 15, 30, 68]. Moreover, some works infuse domain knowledge in finding proper augmentations [44, 41], or design advanced augmentations from the spectral perspective [10, 75, 28, 29]. There are also works identifying limitations in existing task-irrelevant graph augmentations, and expect better practices in augmentations considering the graph-domain knowledge [47, 48, 52]. However, our works are not intended to design stronger data augmentations, but we go another way that finding simple augmentations like random Gaussian noise also work on real-world graph datasets in GCL, while it causes a steep performance drop in VCL.

**Inductive Bias of Architecture to Contrastive Learning.** For the inductive bias of architecture, Saunshi et al. [37] proposes a general theoretical framework showing the importance of architecture inductive bias to standard contrastive learning. Trivedi et al. [48] presents comparable performances between GCL and untrained GCNs on some relatively simple benchmarks, showing the existence of inductive bias of GCL. In contrast, our work firstly uncovers what exactly the inductive bias of GNNs is by exploring the dynamic interplay between the GNN architecture and the contrastive optimization objective during training.

## 3 Preliminaries

Let $\mathcal{G} = (\mathcal{V}, \mathcal{E})$ be a graph with $n$ nodes, where $\mathcal{V}$ and $\mathcal{E}$ are the node set and edge set. We denote $\mathbf{X} \in \mathbb{R}^{n \times h}$ as the node attribute matrix with input dimension $h$ and $\mathbf{A} \in \mathbb{R}^{n \times n}$ as the adjacency matrix. Given augmentation functions $\tau_1, \tau_2 \in \Gamma$, where $\Gamma$ is the set of all possible augmentations defined on graphs, GCL generates two augmented views $\tilde{\mathcal{G}}_1 = \tau_1(\mathcal{G})$ and $\tilde{\mathcal{G}}_2 = \tau_2(\mathcal{G})$. These

augmented views are then embedded into representations via an encoder $f$, which is often followed by a projection head $g$. We denote the backbone features output by the encoder as $\mathbf{H} = f(\mathbf{X}) \in \mathbb{R}^{n \times m}$, and the projection output as $\mathbf{Z} = g(\mathbf{H}) \in \mathbb{R}^{n \times d}$, where the dimension $d$ is often smaller than $m$. During the evaluation, the projection head is removed and only $\mathbf{H}$ is used for downstream tasks. For graph-level tasks, a global representation can be further obtained by applying a pooling operation $r$. The aim of GCL is to learn neural networks that embed a graph into representations by maximizing the representation consistency between different views of the input graph.

For the training objective, the InfoNCE loss [32] is widely used in GCL. Given an anchor view $\mathbf{u}$, which can be a node representation $\mathbf{u} = g(f(\tilde{\mathcal{G}}))_i$ or the global representation of a graph $\mathbf{u} = r(g(f(\tilde{\mathcal{G}})))$, we denote its corresponding positive pair as $\mathbf{v}$. Then, the InfoNCE loss of $\mathbf{u}$ is defined as

$$\mathcal{L}_{\text{InfoNCE}}(\mathbf{u}) = -\log \frac{\exp(s(\mathbf{u}, \mathbf{v})/t)}{\sum_{\mathbf{q} \in \mathcal{N}_{\text{neg}}} \exp(s(\mathbf{u}, \mathbf{q})/t)}, \tag{1}$$

where $\mathcal{N}_{\text{neg}}$ is the set comprising negative samples of $\mathbf{u}$, $s(\cdot, \cdot)$ denotes the cosine similarity, and $t$ is the temperature scale. The InfoNCE loss can be decoupled into two non-overlapping parts, each of which includes only positives or negatives, named as *alignment loss* and *uniformity loss* [53]:

$$\mathcal{L}_{\text{align}}(\mathbf{u}) = -s(\mathbf{u}, \mathbf{v})/t, \quad \mathcal{L}_{\text{uniform}}(\mathbf{u}) = \log \sum_{\mathbf{q} \in \mathcal{N}_{\text{neg}}} \exp(s(\mathbf{u}, \mathbf{q})/t). \tag{2}$$

## 4 How GCL Works without Positive Samples

In this section, we investigate intriguing phenomena of GCL in the absence of positive samples, which are greatly different from the common understanding and practice in VCL. Specifically, we find that positive samples are not necessary for GCL. We highlight the importance of graph convolution in such discrepancy and provide theoretical insights for explaining the success of positive-free GCL. To ensure the validity of our conclusion, we choose seven highly cited GCL methods and evaluate on well-known benchmarks for both node classification and graph classification tasks. Following the convention of GCL, we use the linear-probing protocol for evaluation. We also report the results of the fine-tuning protocol in Appendix K. More details about the adopted methods, experimental settings, and benchmarks can be found in Appendix A. The proof of theorems is attached to Appendix F.

### 4.1 Positive Samples Are NOT a Must in GCL

Upon examining existing approaches like SimCLR [5] and MoCo [17], it becomes evident that positive samples play a vital role in VCL. By maximizing the agreement between positive samples, the neural networks can effectively learn semantic information relevant to downstream tasks [55]. It is widely recognized that without this alignment effect, learned representations may lose meaning and incur poor performance [53, 55]. To illustrate this, we conduct experiments on the CIFAR-10 dataset [25], comparing the InfoNCE loss (including positive samples) and the uniformity loss (excluding positive samples). As shown in Table 1, optimizing only the uniformity loss significantly degrades performance.

Considering the practice in VCL, one would naturally assume that positive samples are equally important and necessary in GCL. However, we unexpectedly find that many existing GCL methods achieve decent performance even without using any positive pairs. To demonstrate this, we conduct comprehensive experiments on both node classification and graph classification tasks. As shown in Table 2, the accuracy gap between the contrastive loss (*Contrast*) and the loss without positives (*NO Pos*) is relatively narrow across most node classification datasets. Similarly, in Table 3, we observe similar phenomena in graph classification, where using loss without positive samples sometimes even outperforms the contrastive loss. Additionally, we perform experiments on randomly initialized

Table 1: Linear probing accuracy (%) of VCL with SimCLR on CIFAR-10: InfoNCE loss, uniformity loss (no positives), alignment loss (no negatives) and no training (random initialization).

| Loss | Accuracy (%) |
|------|------|
| NO Training | $27.20 \pm 0.9$ |
| InfoNCE | $83.51 \pm 0.3$ |
| Uniformity | $27.51 \pm 0.5$ |
| Alignment | $29.67 \pm 0.8$ |

Table 2: Test accuracy (%) of node classification benchmarks using GCL methods. We compare the performances of models trained with InfoNCE loss (Contrast), uniformity loss (NO Pos), alignment loss (NO Neg), and no optimization objective (NO Training). Mean accuracy with standard derivation is reported after 10 runs. Average accuracy across datasets is reported. We conduct significance testing using Wilcoxon Signed Rank Test [57], comparing the contrastive loss and other loss types. The p-value is averaged across datasets. A value below 0.05 denotes significant accuracy difference ( red ), while a value above 0.05 denotes insignificance ( green ). OOM denotes out of memory.

| Method | Loss | Cora | CiteSeer | PubMed | Photo | Computers | Avg | Avg p-value |
|---|---|---|---|---|---|---|---|---|
| GRACE [77] | Contrast | $84.67 \pm 1.39$ | $73.47 \pm 2.32$ | $85.80 \pm 0.16$ | $91.42 \pm 1.27$ | $89.01 \pm 0.60$ | 84.87 | - |
| | NO Training | $69.12 \pm 4.18$ | $60.60 \pm 2.59$ | $80.65 \pm 0.80$ | $68.37 \pm 3.76$ | $57.02 \pm 1.93$ | 67.15 | 0.0020 |
| | NO Pos | $82.65 \pm 1.18$ | $73.50 \pm 2.41$ | $85.28 \pm 0.79$ | $91.32 \pm 1.10$ | $84.40 \pm 0.43$ | 83.43 | 0.2270 |
| | NO Neg | $29.85 \pm 1.45$ | $20.42 \pm 2.26$ | $39.63 \pm 0.81$ | $25.10 \pm 1.74$ | $36.84 \pm 1.30$ | 30.37 | 0.0020 |
| GCA [79] | Contrast | $84.04 \pm 1.55$ | $72.63 \pm 2.68$ | $85.92 \pm 0.69$ | $93.07 \pm 0.66$ | $86.58 \pm 0.75$ | 84.45 | - |
| | NO Training | $71.25 \pm 2.32$ | $58.50 \pm 1.32$ | $80.07 \pm 0.47$ | $84.92 \pm 1.60$ | $68.33 \pm 1.23$ | 72.61 | 0.0020 |
| | NO Pos | $83.09 \pm 2.03$ | $70.42 \pm 3.07$ | $84.68 \pm 0.63$ | $91.50 \pm 0.26$ | $85.19 \pm 0.93$ | 82.98 | 0.1465 |
| | NO Neg | $31.40 \pm 3.61$ | $22.16 \pm 3.01$ | $39.58 \pm 0.83$ | $28.13 \pm 1.14$ | $37.34 \pm 0.95$ | 31.72 | 0.0020 |
| ProGCL [59] | Contrast | $85.42 \pm 3.41$ | $72.85 \pm 2.99$ | OOM | $93.81 \pm 0.48$ | $86.35 \pm 1.28$ | 84.61 | - |
| | NO Training | $79.41 \pm 0.90$ | $58.08 \pm 1.27$ | $83.54 \pm 0.83$ | $84.84 \pm 1.98$ | $68.39 \pm 1.49$ | 74.85 | 0.0023 |
| | NO Pos | $86.76 \pm 0.52$ | $70.76 \pm 1.63$ | OOM | $92.59 \pm 0.16$ | $85.71 \pm 1.32$ | 83.96 | 0.2523 |
| | NO Neg | $30.15 \pm 2.70$ | $21.08 \pm 1.45$ | $21.13 \pm 1.20$ | $4.88 \pm 0.33$ | $3.11 \pm 0.65$ | 16.07 | 0.0020 |

Table 3: Test accuracy (%) of graph classification benchmarks using GCL methods. We compare the performances of models trained with InfoNCE loss (Contrast), uniformity loss (NO Pos), alignment loss (NO Neg), and no optimization objective (NO Training). Mean accuracy with standard derivation is reported after 10 runs. Average accuracy across datasets is reported. We conduct significance testing using Wilcoxon Signed Rank Test [57], comparing the contrastive loss with other loss types. The p-value is averaged across datasets. A value below 0.05 denotes significant accuracy difference ( red ), while a value above 0.05 indicates insignificance ( green ).

| Method | Loss | MUTAG | PTC-MR | PROTEINS | IMDB-B | IMDB-M | REDDIT-B | Avg | Avg p-value |
|---|---|---|---|---|---|---|---|---|---|
| GraphCL [66] | Contrast | $86.36 \pm 1.74$ | $61.73 \pm 1.40$ | $72.98 \pm 0.52$ | $71.96 \pm 0.29$ | $49.80 \pm 0.23$ | $84.92 \pm 0.40$ | 71.29 | - |
| | NO Training | $80.85 \pm 2.99$ | $57.60 \pm 0.66$ | $56.97 \pm 4.08$ | $59.24 \pm 1.64$ | $34.65 \pm 0.67$ | $80.05 \pm 0.35$ | 61.56 | 0.0039 |
| | NO Pos | $87.97 \pm 1.85$ | $62.27 \pm 1.29$ | $73.44 \pm 0.97$ | $72.38 \pm 0.83$ | $48.72 \pm 0.68$ | $82.05 \pm 0.89$ | 71.14 | 0.3281 |
| | NO Neg | $88.73 \pm 0.52$ | $58.03 \pm 2.24$ | $73.60 \pm 0.79$ | $72.10 \pm 0.32$ | $49.61 \pm 0.33$ | $82.56 \pm 0.76$ | 70.77 | 0.1650 |
| ADGCL [45] | Contrast | $90.43 \pm 1.18$ | $57.05 \pm 2.58$ | $74.29 \pm 1.02$ | $71.96 \pm 0.15$ | $50.16 \pm 0.17$ | $84.84 \pm 0.45$ | 71.46 | - |
| | NO Training | $59.99 \pm 0.43$ | $53.80 \pm 0.82$ | $55.26 \pm 4.30$ | $50.98 \pm 0.91$ | $33.55 \pm 0.46$ | $59.08 \pm 3.49$ | 52.11 | 0.0026 |
| | NO Pos | $89.47 \pm 0.77$ | $57.54 \pm 1.93$ | $72.81 \pm 0.63$ | $71.72 \pm 0.27$ | $49.89 \pm 0.46$ | $84.35 \pm 0.49$ | 70.96 | 0.2419 |
| | NO Neg | $88.51 \pm 0.85$ | $56.14 \pm 2.49$ | $73.98 \pm 0.32$ | $71.74 \pm 0.12$ | $48.93 \pm 0.33$ | $74.81 \pm 0.35$ | 69.02 | 0.1631 |
| JOAO [67] | Contrast | $86.17 \pm 1.55$ | $61.47 \pm 1.53$ | $73.15 \pm 0.92$ | $71.86 \pm 0.32$ | $48.80 \pm 0.52$ | $82.51 \pm 0.87$ | 70.66 | - |
| | NO Training | $78.54 \pm 2.76$ | $55.48 \pm 1.85$ | $56.78 \pm 2.91$ | $55.08 \pm 1.55$ | $34.93 \pm 0.67$ | $52.35 \pm 1.19$ | 55.53 | 0.0020 |
| | NO Pos | $85.42 \pm 1.33$ | $60.36 \pm 2.19$ | $73.98 \pm 0.53$ | $71.06 \pm 0.52$ | $48.11 \pm 0.61$ | $83.14 \pm 0.34$ | 70.35 | 0.3057 |
| | NO Neg | $85.19 \pm 0.84$ | $58.13 \pm 0.92$ | $73.64 \pm 0.73$ | $69.46 \pm 0.52$ | $47.93 \pm 0.56$ | $82.31 \pm 1.47$ | 69.44 | 0.1211 |
| InfoGraph [42] | Contrast | $89.75 \pm 1.35$ | $64.26 \pm 0.30$ | $72.22 \pm 0.51$ | $72.04 \pm 0.54$ | $49.49 \pm 0.31$ | $82.46 \pm 0.52$ | 71.70 | - |
| | NO Training | $85.63 \pm 0.32$ | $54.84 \pm 1.56$ | $56.93 \pm 0.30$ | $58.44 \pm 1.66$ | $35.79 \pm 0.81$ | $56.61 \pm 2.07$ | 58.04 | 0.0020 |
| | NO Pos | $88.58 \pm 1.49$ | $62.28 \pm 1.34$ | $70.58 \pm 0.54$ | $73.98 \pm 0.60$ | $49.37 \pm 0.54$ | $81.55 \pm 0.88$ | 71.06 | 0.2975 |
| | NO Neg | $88.19 \pm 0.90$ | $62.23 \pm 1.62$ | $68.88 \pm 0.98$ | $72.34 \pm 0.66$ | $49.19 \pm 0.63$ | $80.46 \pm 0.87$ | 70.22 | 0.1680 |
| AutoGCL [64] | Contrast | $86.26 \pm 1.14$ | $61.67 \pm 0.60$ | $67.44 \pm 1.16$ | $72.50 \pm 0.68$ | $49.89 \pm 0.45$ | $81.43 \pm 1.89$ | 69.87 | - |
| | NO Training | $85.80 \pm 4.25$ | $56.27 \pm 3.86$ | $53.98 \pm 4.40$ | $57.96 \pm 0.76$ | $34.61 \pm 0.82$ | $66.30 \pm 0.05$ | 59.15 | 0.0052 |
| | NO Pos | $87.23 \pm 1.13$ | $59.15 \pm 0.76$ | $66.57 \pm 1.53$ | $70.60 \pm 1.13$ | $49.31 \pm 0.52$ | $79.61 \pm 1.05$ | 68.75 | 0.1566 |
| | NO Neg | $87.12 \pm 1.71$ | $60.35 \pm 1.14$ | $70.51 \pm 1.18$ | $73.20 \pm 0.32$ | $50.08 \pm 0.54$ | $82.33 \pm 0.90$ | 70.60 | 0.1403 |

models without training (*NO Training*) for comparison, which result in poor representations. These findings suggest that the removal of positive samples in GCL has minimal impact on the performance of downstream benchmarks, in stark contrast to the results in VCL (Table 1). For further illustration, we visualize the representations learned with contrastive loss and uniformity loss using T-SNE [49] in Appendix B. We also conduct extensive experiments on heteophily datasets and large benchmark OGB-arxiv [20] (See Appendix C).

Table 4: Test accuracy (%) of node classification benchmarks using GRACE method with MLP encoder. We compare the performances of models trained with InfoNCE loss (Contrast), uniformity loss (NO Pos), alignment loss (NO Neg), and no training objective (NO Training). Mean accuracy with standard derivation is reported after 10 runs. Average accuracy across datasets is reported. We conduct significance testing using Wilcoxon Signed Rank Test [57], comparing the contrastive loss with other loss types. The p-value is averaged across datasets. A value below 0.05 denotes a significant accuracy difference ( red ), while a value above 0.05 indicates insignificance ( green ).

| Method | Loss | Encoder | Cora | CiteSeer | PubMed | Photo | Computers | Avg | Avg p-value |
|--------|------|---------|------|----------|--------|-------|-----------|-----|-------------|
| GRACE [77] | Contrast | MLP | $67.72 \pm 0.88$ | $65.51 \pm 2.63$ | $83.29 \pm 0.49$ | $87.92 \pm 0.59$ | $80.89 \pm 1.21$ | 77.07 | - |
| | NO Training | MLP | $40.66 \pm 2.49$ | $42.81 \pm 4.82$ | $78.53 \pm 0.90$ | $62.12 \pm 0.97$ | $57.97 \pm 1.13$ | 56.42 | 0.0020 |
| | NO Pos | MLP | $56.10 \pm 1.08$ | $49.82 \pm 3.76$ | $81.32 \pm 0.77$ | $65.25 \pm 1.13$ | $61.37 \pm 0.74$ | 62.77 | 0.0020 |
| | NO Neg | MLP | $51.69 \pm 3.04$ | $50.36 \pm 1.14$ | $79.00 \pm 0.63$ | $61.33 \pm 0.75$ | $55.07 \pm 0.99$ | 59.49 | 0.0020 |

## 4.2 The Implicit Regularization of Graph Convolution in GCL

The intriguing property of positive samples in GCL encourages us to explore the underlying reasons behind this phenomenon. It is worth noting that all the GCL methods analyzed above adopt message-passing graph neural networks (GNNs) like GCN [24] as backbone encoders. We aim to demonstrate that these GNNs inherently possess an implicit regularization effect that facilitates the aggregation of positive samples. This finding helps elucidate why GCL can achieve satisfactory performance without explicitly incorporating an alignment objective.

To simplify the explanation, we focus on the vanilla graph convolution module proposed in GCN as an illustrative example. At the $l$-th layer, node representations are aggregated through two interleaving steps:

$$(\text{Graph Convolution}) \ \mathbf{H}' = \hat{\mathbf{A}}\mathbf{H}^{(l)}, \tag{3}$$

$$(\text{Feature Transformation}) \ \mathbf{H}^{(l+1)} = \sigma(\mathbf{H}'\mathbf{W}^{(l)}), \tag{4}$$

where $\sigma$ is the activation function, and $\mathbf{W}^{(l)}$ denotes the weight matrix. $\hat{\mathbf{A}} = \bar{\mathbf{D}}^{-1/2}\bar{\mathbf{A}}\bar{\mathbf{D}}^{-1/2}$ is the symmetrically normalized version of the self-loop augmented adjacency matrix $\bar{\mathbf{A}} = \mathbf{A} + \mathbf{I}$, where $\bar{\mathbf{D}}$ is the diagonal degree matrix of $\bar{\mathbf{A}}$. While various variants of GCN have been proposed, most include generalized graph convolution (GraphConv) operators that bring neighbor features closer through message passing [50]. For comparison, we also consider a vanilla MLP encoder, which extracts features from each node individually and can be seen as only applying the feature transformation step.

Importantly, we notice that the utilization of GraphConv within encoders is the key for most existing GCL methods to generalize well in the absence of positive samples. To demonstrate this, we compare two backbones: GCN (with GraphConv) and MLP (without GraphConv), on the node-classification task. The results are presented in Table 4. Remarkably, we observe that the distinctive property of GCL disappears under the MLP backbone. Compared to the performance achieved with the standard contrastive loss (*e.g.,* 67.72% on Cora), the MLP-based GCL exhibits significantly lower performance (56.10%) under the no-positive setting. This highlights the importance of the feature propagation process in GraphConv, which underlies the unique positive-free behavior observed in GCL.

Here we provide theoretical insights into this phenomenon by uncovering the implicit regularization mechanism of GraphConv in graph contrastive learning. Through formal connections established between GraphConv and a neighbor-induced alignment objective, we demonstrate that GraphConv has the capability to replace the positive alignment loss in GCL. Consequently, GCL attains favorable performance even in the absence of explicit positive sample training.

**Theorem 4.1.** *Suppose the positive node pairs $(x, x^+) \sim P_\mathcal{G}$ is drawn from the following distribution defined via the normalized connection weight of the graph $\mathcal{G}$*

$$\mathcal{P}_\mathcal{G}(x, x^+) = \frac{\hat{\mathbf{A}}_{x,x^+}}{\sum_{u,v} \hat{\mathbf{A}}_{u,v}}, \quad \forall x, x' \in [N], \tag{5}$$

*then a step of GraphConv (Eq 3) will decrease the following feature alignment loss between positive samples (here $\mathbf{h}_x$ refers to the $x$-th row of a feature matrix $\mathbf{H}$):*

$$\tilde{\mathcal{L}}_{\text{align}}(\mathbf{h}_x) = -\mathbb{E}_{x,x^+ \sim \mathcal{P}_\mathcal{G}(x,x^+)}[\mathbf{h}_x^\top \mathbf{h}_{x^+}]. \tag{6}$$

Theorem 4.1 reveals that GraphConv implicitly achieves feature alignment among neighborhood samples. This alignment process is particularly effective in homophilic graphs, where neighbors predominantly belong to the same class. In this context, the neighbors essentially act as high-quality positive samples. As a result, the neighbor-induced alignment loss can effectively cluster intra-class samples together, providing a plausible explanation for the success of positive-free GCL. Interestingly, the connection between graph convolution and the alignment objective can also provide a natural explanation for why Yang et al. [63] works, which applies graph convolution to a trained MLP and observes improved performance. This strategy, from our perspective, is amount to further training the MLP features with a neighbor-induced alignment loss for a few steps (thus no severe feature collapse), thus helps improve MLP's performance.

# 5 How GCL Works without Negative Samples

In this section, we investigate intriguing phenomena of GCL in the absence of negative samples. There are works showing contrastive learning can get rid of negative samples by specific designs on architectures [46, 80] or objective functions [71, 2]. However, we observe that negative samples are dispensable without any specific designs for the graph classification task, whereas in the node classification task, simply removing them may not be sufficient. From the perspective of feature collapse, we emphasize the significant role played by the projection head in the graph classification task. Building upon this insight, we address the collapse issue in the node classification task by modifying the backbone encoder with a specialized normalization technique, and further give a theoretical explanation. The experimental settings are identical to those in Section 4, and the proof of theorems is attached to Appendix F.

## 5.1 Graph Classification: Both Negative Samples and Specific Designs Are Not Needed

In the context of VCL, it is widely recognized that the removal of negative samples alone leads to the failure of methods. This is primarily attributed to the fact that without negative samples, the alignment loss can be easily minimized by adopting a shortcut solution where the encoder generates a constant feature for all samples, *i.e.,* $f(x) = c, \forall\, x$. As a consequence, this collapsed feature lacks discriminative power for downstream tasks, resulting in a phenomenon referred to as feature collapse [21]. To empirically demonstrate this issue, we present the evident performance degradation observed in pure no-negative VCL in Table 1.

In contrast to the VCL scenario, our findings in GCL for the graph classification task reveal a notable difference. Surprisingly, we discover that GCL can perform well by utilizing the vanilla positive alignment loss alone, without the need for negative samples or any modifications to the network architecture. As demonstrated in Table 3, models trained exclusively with positive pairs achieve comparable or even superior performance compared to those trained with the default contrastive loss.

Further investigation into the architecture reveals the intriguing role of the projection head in the no-negative setting. Specifically, we estimate the average similarity of the representations $\mathbf{H} = f(\mathbf{X})$ output by the encoder and $\mathbf{Z} = g(\mathbf{H})$ output by the projection head (Figure 1(a)). The similarity of $\mathbf{Z}$ is close to 1, indicating that the projection head indeed learns a collapsed solution. However, the similarity of $\mathbf{H}$ is much lower. It is worth noting that in common GCL practice, the projection head is removed after training, and downstream tasks employ $\mathbf{H}$ for evaluation. Therefore, thanks to the projection head, although the model learns a collapsed solution for optimizing the alignment loss, the downstream results remain unaffected.

To gain further insights into the role of the projection head, we explore the mechanism from a spectral perspective. We compare the singular value distributions of representations before and after the projection head on the MUTAG dataset (Figure 1(b)). The distribution of singular value becomes more concentrated after the projection head, indicating a decrease in rank. To validate this, we estimate the ranks of $\mathbf{H}$ and $\mathbf{Z}$, revealing that the rank of $\mathbf{Z}$ is noticeably lower than that of $\mathbf{H}$ (Figure 1(c)). Inspired by Gupta et al. [13], we remove the projection head during training (where $g(\cdot)$ is an identity function, making $\mathbf{H}$ equal to $\mathbf{Z}$). Comparing the rank of resulting representation $\mathbf{H}^*$, we find that it falls between the ranks of $\mathbf{H}$ and $\mathbf{Z}$. These observations indicate that the projection head implicitly selects a low-rank subspace of features to satisfy the alignment loss.

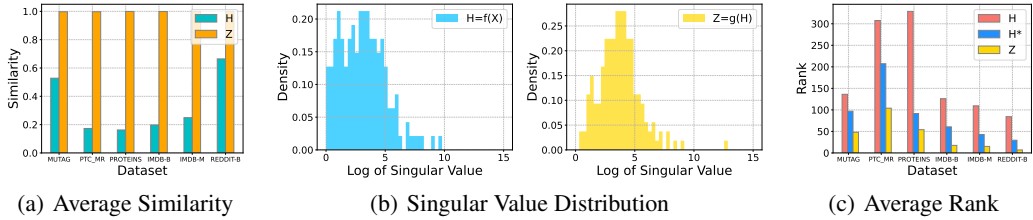

(a) Average Similarity      (b) Singular Value Distribution      (c) Average Rank

Figure 1: Metrics before and after the projection head. $\mathbf{H}$ and $\mathbf{Z}$ denote the representations before and after the projection head, respectively. In Figure 1(c), we also report the rank of representations $\mathbf{H}^*$ when training without the projection head. Experiments are conducted with GraphCL method.

Table 5: Test accuracy (%) of node classification benchmarks using GRACE method. We compare the performances of models trained with InfoNCE loss (Contrast), alignment loss (NO Neg), and alignment loss with ContraNorm in encoder (GCN+CN). Mean accuracy with standard derivation is reported after 10 runs. Average accuracy across datasets is reported. We conduct significance testing using Wilcoxon Signed Rank Test [57], comparing the default setting (first line) with others. The p-value is averaged across datasets. A value below 0.05 denotes significant accuracy difference ( red ), while a value above 0.05 indicates insignificance ( green ).

| Method | Loss | Encoder | Cora | CiteSeer | PubMed | Photo | Computers | Avg | Avg p-value |
|--------|------|---------|------|----------|--------|-------|-----------|-----|-------------|
| | Contrast | GCN | $84.67 \pm 1.39$ | $73.47 \pm 2.32$ | $85.80 \pm 0.16$ | $91.42 \pm 1.27$ | $89.01 \pm 0.60$ | 84.87 | - |
| GRACE [77] | NO Neg | GCN | $29.85 \pm 1.45$ | $20.42 \pm 2.26$ | $39.63 \pm 0.81$ | $25.10 \pm 1.74$ | $36.84 \pm 1.30$ | 30.37 | 0.0020 |
| | NO Neg | GCN + CN | $82.35 \pm 2.28$ | $72.25 \pm 1.86$ | $83.30 \pm 0.63$ | $92.43 \pm 0.82$ | $84.48 \pm 1.01$ | 82.96 | 0.1621 |

## 5.2 Node Classification: Normalization in the Encoder Is Enough

When it comes to the node classification task, the absence of negative samples in GCL leads to a significant performance drop (Table 2), unlike the case in graph classification. Numerous factors may be responsible for the suboptimal performance. To confirm feature collapse is the underlying cause, we visualize the training process with alignment loss, where the average node similarities of $\mathbf{H}$ and $\mathbf{Z}$ both unite towards one at the end of training (see Appendix D).

Notably, different from graph classification, the representations learned by the encoder also collapse in the node classification task. One plausible conjecture is that learning a collapsed solution is relatively easier for the global graph representation, which can be achieved solely by the projection head. In this case, the encoder is preserved from collapse. However, for learning local node representations, the alignment loss requires each node to be collapsed, which often needs the encoder's involvement. We provide empirical insights in Appendix E, while a rigorous theoretical understanding remains a topic for future work.

Now, the question arises: how does GCL manage to work without negative samples for node classification? While previous solutions derived from VCL [11, 6, 69, 1], such as asymmetric architectures [46, 22] or feature decorrelation objectives [2, 71], exist, they are specific designs which cannot be easily generalized to other methods. Recalling that the feature collapse can be traced back to the encoder, a straightforward approach is directly changing the encoder to prevent collapse. Specifically, we find that just incorporating a normalization component called ContraNorm (CN) [12] into the encoder of GCL is enough.

ContraNorm is originally designed for alleviating the over-smoothing problem in GNNs and Transformers with the formulation:

$$\mathrm{CN}(\mathbf{H}) = \mathbf{H} - \alpha \mathbf{D} \tilde{\mathbf{A}} \mathbf{H}. \tag{7}$$

Here, $\mathbf{H}$ is hidden representation, and $\alpha$ is a hyper-parameter for scaling. $\mathbf{D}$ is the diagonal degree matrix of $\tilde{\mathbf{A}}$, where $\tilde{\mathbf{A}} = \mathrm{softmax}(\mathbf{D} \mathbf{H} \mathbf{H}^\top)$ computes the row-wise normalized similarity matrix between node features. Here, we use a degree-normalized variant of ContraNorm and are the first to introduce it as a novel extension to GCL. As seen from Table 5, by simply incorporating the normalization layer into the encoder, the collapse issue can be rooted out for the GRACE method. Importantly, this normalization layer can be easily adapted by other GCL methods in a plug-and-play

manner. We validate the effectiveness of ContraNorm on multiple GCL methods and under different encoders. The results are provided in Appendix H due to space constraints. It is worth noting that our proposed approach maintains a symmetric model architecture with only the alignment loss, highlighting the ability of the normalization in the encoder for no-negative GCL to perform well.

### 5.3 ContraNorm Performs Negative Uniformity Implicitly

In this part, we explain how ContraNorm prevents feature collapse without other special designs in the no-negative setting. Similar to the GraphConv case, we define a uniformity loss among all nodes in the same graph to promote feature diversity, namely the neighbor-induced uniformity loss. The following theorem proves that the update of ContraNorm layer leads to a decrease in this uniformity loss.

**Theorem 5.1.** *Suppose the sample $x$ is drawn from the same distribution in Theorem 4.1, the neighbor-induced uniformity loss is defined as*

$$\tilde{\mathcal{L}}_{\text{uniform}} = \mathbb{E}_{x \sim P_{\mathcal{G}}(x)}[\log \mathbb{E}_{x' \sim P_{\mathcal{G}}(x)} \exp(\mathbf{h}_x^\top \mathbf{h}_{x'})]. \tag{8}$$

*The gradient update of this uniformity loss with step size $\alpha > 0$ gives the ContraNorm update (Eq. 7).*

The derived update discussed in Guo et al. [12] suggests that the ContraNorm layer can implicitly promote the diversity among node features during the propagation process. This explains why combining GCL with ContraNorm can avoid feature collapse without explicitly relying on any negative samples.

By analyzing the roles of GraphConv and ContraNorm, we notice that in contrast to visual contrastive learning where the encoders primarily extract features from individual samples, the feature propagation layers in GNNs (GraphConv and ContraNorm) also capture the interaction between different samples. This property enables them to effectively replace the roles of inter-sample objectives like the alignment and uniformity losses. In other words, the feature propagation layers inherently encode the necessary information for learning meaningful representations without explicitly relying on inter-sample objectives. Specifically, by combining Theorem 4.1 and Theorem 5.1, we show that the joint update using graph convolution and ContraNorm implicitly optimizes a neighbor-induced contrastive learning loss:

**Theorem 5.2.** *The joint update of GraphConv and ContraNorm, i.e.,*

$$\mathbf{H}_{\text{new}} = (\mathbf{I} + \hat{\mathbf{A}})\mathbf{H} - \alpha \mathbf{D}\tilde{\mathbf{A}}\mathbf{H} \tag{9}$$

*corresponds to a gradient descent update of the following contrastive learning loss:*

$$\begin{aligned}
\tilde{\mathcal{L}}_{\text{contrast}} &= \tilde{\mathcal{L}}_{\text{align}} + \tilde{\mathcal{L}}_{\text{uniform}} \\
&= \mathbb{E}_{x,x^+ \sim \mathcal{P}_{\mathcal{G}}(x,x^+)}[\mathbf{h}_x^\top \mathbf{h}_{x^+}] + \mathbb{E}_{x \sim P_{\mathcal{G}}(x)}[\log \mathbb{E}_{x' \sim P_{\mathcal{G}}(x)} \exp(\mathbf{h}_x^\top \mathbf{h}_{x'})].
\end{aligned} \tag{10}$$

## 6 Simple Augmentations Do Not Destroy GCL Performance

Data augmentation is the arguably most crucial component of VCL methods, since different kinds of data augmentations have a dramatic influence on its final performance. When examining Table 6, we observe a substantial degradation ($83.51\% \to \approx 30\%$) in VCL's performance when removing all augmentations or applying only random Gaussian noise. However, we find that data augmentations have a much smaller influence on GCL methods.

In GCL, basic augmentations are typically domain-specific, tailored to the node features and topology information, *e.g.,* node feature masking, edge perturbation, and subgraph sampling. Here we choose the combination of node feature masking (FM) and edge perturbation (EP) as the baseline augmentation, which is widely adopted in GCL methods [77, 79, 46, 71].

Table 6: Linear probing accuracy (%) of VCL with SimCLR on CIFAR-10 with different augmentation settings.

| Augentation | Accuracy(%) |
|---|---|
| NO Aug | $28.29 \pm 1.0$ |
| Default Aug | $83.51 \pm 0.3$ |
| Gaussian | $36.56 \pm 1.2$ |

Taking the node classification as an example, following common practices when unknowing the data prior [33, 70], we further consider a simple augmentation: random Gaussian noise, where a random noise sample drawn from a Gaussian is directly added to node features. Formally, given a graph $\mathcal{G} = (\mathbf{A}, \mathbf{X})$, the random noise augmentation is defined as $\tau(\mathcal{G}) = (\mathbf{A}, \mathbf{X} + \varepsilon), \varepsilon \sim \mathcal{N}(0, \sigma^2)$. In practice, we select the standard deviation $\sigma$ selected from $[1e-4, 5e-4, 1e-5]$. For comparison, we also include the no augmentation setting. The results are shown in Table 7. Although independent of the graph structure and node attributes, the random noise augmentation still achieves a comparable performance compated to domain-specific augmentations, which is quite different from the observations in VCL (Table 6). For further verification, we also conduct experiments with MLP and under different loss settings, the details are shown in Appendix I.

The robustness of GCL to random noise augmentations highlights its flexibility and resilience in the absence of domain-specific augmentations. Recalling Section 4, graph contrastive learning is equipped with two kinds of alignment properties: alignment loss and graph convolution. Therefore, when the effect of alignment loss is weakened (corresponding to domain-agnostic augmentations) or even removed (corresponding to no augmentations), the performance of GCL is relatively slightly influenced with the graph convolution as backing.

Table 7: Test accuracy (%) of node classification benchmarks using GRACE with different augmentations. We compare no augmentations (NO Aug), domain-agnostic augmentations (Gaussian), and default domain-specific augmentations (FM+EP). Average accuracy and p-value are reported.

| Augmentation | Cora | CiteSeer | PubMed | Photo | Computers | Avg | Avg p-value |
|---|---|---|---|---|---|---|---|
| NO Aug | $79.56 \pm 2.18$ | $71.83 \pm 1.83$ | $84.68 \pm 0.58$ | $90.99 \pm 1.26$ | $82.83 \pm 0.86$ | 81.98 | 0.1051 |
| FM+EP | $84.67 \pm 1.39$ | $73.47 \pm 2.32$ | $85.80 \pm 0.16$ | $91.42 \pm 1.27$ | $89.01 \pm 0.60$ | 84.87 | - |
| Gaussian | $82.72 \pm 2.38$ | $72.60 \pm 1.21$ | $85.24 \pm 0.61$ | $91.32 \pm 1.37$ | $82.87 \pm 1.09$ | 82.95 | 0.1778 |

# 7 Discussion and Conclusion

In this paper, we have shown that GCL exhibits many intriguing phenomena that are rather contradictory to those in VCL. Specifically, we have found that GCL can work in the absence of positive samples. Second, GCL works well without negative pairs for the graph classification task. Third, GCL can achieve comparable performance with domain-agnostic data augmentations like random Gaussian noise. We have made these observations via extensive experiments with a wide range of representative GCL methods. However, the empirical experiments cannot cover all of the GCL methods. We indeed find some exceptions and give a concrete discussion in Appendix G, where exceptional observations can be attributed to the individual property of methods.

Notably, we highlight the implicit mechanisms of architectures to contrastive learning. Theoretically, we build the connection between graph convolution and a neighbor-induced alignment loss, as well as the connection between ContraNorm and a neighbor-induced uniformity loss, giving explanations for the above unique properties of GCL. Overall, our method suggests that graph contrastive learning may behave quite differently from its visual counterpart, and more efforts should be brought in for designing graph-specific self-supervised learning.

Since the main goal of this work is to examine the roles of each component of GCL objectives, one limitation is that it does not propose a new GCL method. Nevertheless, we believe that the new findings in this work would be valuable for future GCL designs. Also, the paper does not examine other SSL paradigms on graph, like masked modeling, which would be an interesting direction to explore in the future.

## Acknowledgements

Yisen Wang was supported by National Key R&D Program of China (2022ZD0160304), National Natural Science Foundation of China (62006153, 62376010, 92370129), Open Research Projects of Zhejiang Lab (No. 2022RC0AB05), and Beijing Nova Program (20230484344).

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

# A  Details on GCL Methods, Benchmarks and Experiment Settings

## A.1  Brief Introduction of GCL Methods

**Methods for the node classification task.**

- **GRACE** [77]. GRACE generates two graph views by corruption and learns node representations by maximizing the agreement of node representations in these two views. To provide diverse node contexts for the contrastive objective, GRACE proposes a hybrid scheme for generating graph views on both structure and attribute levels.

- **GCA** [79]. GCA proposes adaptive augmentation that incorporates various priors for topological and semantic aspects of the graph. On the topology level, GCA designs augmentation schemes based on node centrality measures, while on the node attribute level, GCA corrupts node features by adding more noise to unimportant node features.

- **ProGCL** [59]. ProGCL observes limited benefits when adopting existing hard negative mining techniques of other domains in graph contrastive learning. ProGCL proposes an effective method to estimate the probability of a negative being true one, and devises two schemes to boost the performance of GCL.

- **DGI** [51]. DGI relies on maximizing mutual information between patch representations and corresponding high-level summaries of graphs—both derived using established graph convolutional network architectures. The learnt patch representations summarize subgraphs centered around nodes of interest, and can thus be reused for downstream node-wise learning tasks.

- **MVGRL** [14]. MVGRL introduces a self-supervised approach for learning node and graph level representations by contrasting structural views of graphs. MVGRL shows that unlike visual representation learning, increasing the number of views to more than two or contrasting multi-scale encodings does not improve performance, and the best performance is achieved by contrasting encodings from first-order neighbors and graph diffusion.

**Methods for the graph classification task.**

- **GraphCL** [66]. GraphCL designs four types of graph augmentations to incorporate various priors, and learns graph-level representations by maximizing the global representations of two views for a graph.

- **ADGCL** [45]. ADGCL proposes a novel principle, adversarial GCL, which enables GNNs to avoid capturing redundant information during training by optimizing adversarial graph augmentation strategies used in GCL.

- **JOAO** [67]. JOAO proposes a unified bi-level optimization framework to automatically, adaptively and dynamically select data augmentations when performing GraphCL on specific graph data. JOAO is instantiated as min-max optimization.

- **InfoGraph** [42]. InfoGraph maximizes the mutual information between the graph-level representation and the representations of substructures of different scales (*e.g.,* nodes, edges, triangles). By doing so, the graph-level representations encode aspects of the data that are shared across different scales of substructures.

- **AutoGCL** [64]. AutoGCL employs a set of learnable graph view generators orchestrated by an auto augmentation strategy. The learnable view generators, the graph encoder, and the classifier are trained jointly in an end-to-end manner.

## A.2  Relation between Our Work and SCE [74]

Inspired by the sparsest cut problem, SCE [74] introduces a Laplacian smoothing trick and then proposes a model without positive samples but only using negative samples for training. It is noted that SCE is a specific node-level GCL method, while we provide a comprehensive comparison for representative GCL methods on a range of datasets. Specifically, the two differ in: 1) Tasks: SCE only considers node classification tasks while we consider both graph and node classification tasks on various datasets. 2) Backbone networks: The backbone of SCE is a special multi-scale GCN variant, which adopts linear graph convolution (like SGC [58]) and aggregates multi-scale features

at last (like JK-Net [60]). In comparison, we adopt GCN for node classification tasks and GIN for graph classification tasks following the common practice. 3) Learning objectives: For training, SCE designs a new formulation of uniformity loss (the inverse of total pairwise distance) ($L_{unsup}$) and an L2 regularization on model weights ($L_2$):

$$\mathcal{L} = \alpha \mathcal{L}_{unsup} + \beta \mathcal{L}_2 = \frac{\alpha}{\sum_{(v_i, v_j) \in \mathcal{N}} \|z_i - z_j\|^2} + \beta \|\theta\|^2.$$

Therefore, SCE's unique backbone and objectives raise questions about the general applicability of their findings. Instead, theoretically and empirically, we demonstrate the general validity of the non-necessity of positive samples across various tasks, backbones, and GCL objectives.

### A.3 Introduction of Graph Benchmarks

**Node classification benchmarks**. 1) Citation Networks [38, 31]. Cora, CiteSeer and PubMed are three popular citation graph datasets. In these graphs, nodes represent papers and edges correspond to the citation relationship between two papers. Nodes are classified according to academic topics. 2) Amazon Co-purchase Networks [39]. Photo and Computers are collected by crawling Amazon websites. Goods are represented as nodes and the co-purchase relationships are denoted as edges. Node features are the bag-of-words representation of product reviews. Each node is labeled with the category of goods. 3) Wikipedia Networks [36]. Squirrel and Chameleon was collected from the English Wikipedia, representing page-page networks on specific topics. Nodes represent articles and edges are mutual links between them.

**Graph Classification benchmarks**. 1) Molecules. MUTAG [8] is a dataset of nitroaromatic compounds and the goal is to predict their mutagenicity on Salmonella typhimurium. PTC-MR [18] is a collection of 344 chemical compounds represented as graphs that report carcinogenicity for male or female rats. 2) Bioinformatics. PROTEINS [3] is a dataset of proteins that are classified as enzymes or non-enzymes. Nodes represent the amino acids and two nodes are connected by an edge if they are less than 6 Angstroms apart. 3) Social Networks. IMDB-BINARY and IMDB-MULTI [62] are movie collaboration datasets consisting of a network of 1,000 actors/actresses who played roles in movies in IMDB. In each graph, nodes represent actors/actresses, and corresponding nodes are connected if they appear in the same movie. REDDIT-BINARY [62] consists of graphs corresponding to online discussions on Reddit. In each graph, nodes represent users, and there is an edge between them if at least one of them responds to the other's comment.

Statistics of datasets are shown in Table 8.

Table 8: Statistics of classification benchmarks. We report average numbers of nodes, edges, and features across graphs in graph classification datasets. For datasets lacking feature attributes, we use all-one vectors as pseudo attributes in practice.

| Task | Category | Dataset | #Graphs | # Nodes | # Edges | # Features | # Classes |
|------|----------|---------|---------|---------|---------|------------|-----------|
| Node | Citation | Cora | 1 | 2,708 | 5,278 | 1,433 | 7 |
| | | CiteSeer | 1 | 3,327 | 4,552 | 3,703 | 6 |
| | | PubMed | 1 | 19,717 | 44,338 | 500 | 3 |
| | Co-purchase | Photo | 1 | 7,650 | 119,081 | 745 | 8 |
| | | Computers | 1 | 13,752 | 245,861 | 767 | 10 |
| | Wikipedia | Chameleon | 1 | 2,277 | 36,101 | 500 | 6 |
| | | Squirrel | 1 | 5,201 | 217,073 | 2,089 | 4 |
| Graph | Protein | MUTAG | 188 | 17.9 | 39.6 | 7 | 2 |
| | | PTC-MR | 344 | 14.3 | 29.4 | 18 | 2 |
| | Bioinformatics | PROTEINS | 1113 | 39.1 | 145.6 | 0 | 2 |
| | Social Networks | IMDB-BINARY | 1000 | 19.8 | 193.1 | 0 | 2 |
| | | IMDB-MULTI | 1500 | 13.0 | 131.9 | 0 | 3 |
| | | REDDIT-BINARY | 2000 | 429.6 | 995.5 | 0 | 2 |

### A.4 Experimental Details

For the node classification task, following Zhu et al. [77], Velickovic et al. [51], Hassani and Khasahmadi [14], we use linear evaluation protocol, where the model is trained in an unsupervised manner and feeds the learned representation into a linear logistic regression classifier. In the training procedure, a 2-layer Graph Convolutional Network (GCN) [24] is adopted as the encoder. We adopt the default settings of Zhu et al. [77]. Specifically, we use removing edges and masking node features as data augmentations. We grid search augmentation ratios in $\{0.0, 0.1, 0.2, 0.3, 0.4\}$. All experiments are trained with Adam SGD optimizer [23] with the learning rate selected from $\{0.01, 0.001, 0.0005\}$. The epoch number is selected from $\{200, 1000, 2000\}$. The other parameters are fixed for all datasets. In the evaluation procedure, we randomly split each dataset with a training ratio of 0.8 and a test ratio of 0.1, and hyperparameters are fixed as the same for all the experiments. Each experiment is repeated ten times with mean and standard derivation of accuracy score.

For the graph classification task, in the training procedure, a Graph Isomorphism Network (GIN) [61] is adopted as the encoder whose layer number is chosen from $\{4, 8, 12\}$ and hidden dimension chosen from $\{32, 512\}$. We use Adam SGD optimizer with the learning rate selected in $\{10^{-3}, 10^{-4}, 10^{-5}\}$ and the number of epochs in $\{20, 100\}$. Following Sun et al. [42], You et al. [66], we feed the generated graph embeddings into a linear Support Vector Machine (SVM) classifier, and the parameters of the downstream classifier are independently tuned by cross-validation. The C parameter is tuned in $\{10^{-3}, 10^{-2}, \cdots, 10^2, 10^3\}$. We report the mean 10-fold cross-validation accuracy with standard deviation. All experiments are conducted on a single 24GB NVIDIA GeForce RTX 3090.

For the image classification task, we pretrain ResNet-18 [16] on the CIFAR-10 dataset for 200 epochs, with a projection dimension of 128 and a batch size of 512. We use the SGD optimizer and cosine annealing schedule to set the learning rate, which is initialized as 0.6. During the fine-tuning phase, we only optimize the linear layer for 100 epochs, using the same learning rate schedule as in the pretrain phase. To evaluate the performance, we report the mean accuracy and standard deviation over 5 independent experiments.

Across our experiments, we follow the standard data splits in each domain. To further resolve concerns on the consistency of evaluation setups, we unify the evaluation settings for GCL and VCL. The detailed results are shown in Appendix L.

## B Visualization of VCL and GCL via T-SNE

To further illustrate the difference between VCL and GCL, we visualize the representations learned with contrastive loss and uniformity loss using T-SNE [49]. The results are shown in Figure 2. For VCL, the representations learned by uniformity loss distribute more randomly without clear decision boundaries, compared to those learned by InfoNCE loss. However, for GCL, the representations learned by the two losses both achieve good clustering effects.

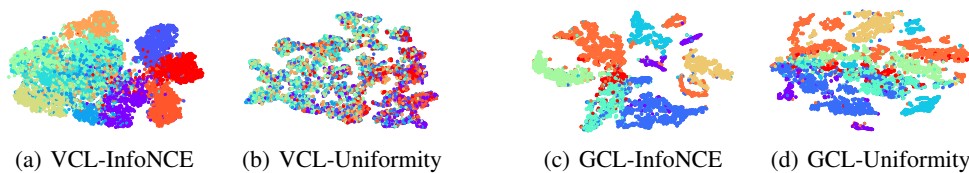

| (a) VCL-InfoNCE | (b) VCL-Uniformity | (c) GCL-InfoNCE | (d) GCL-Uniformity |

Figure 2: T-SNE visualization of representations learned by VCL and GCL, with InfoNCE loss and uniformity loss. Figure 2(a) and 2(b) are conducted with SimCLR on CIFAR10. Figure 2(c) and 2(d) are conducted with GRACE on Amazon-Photo dataset.

## C Results of Extensive benchmarks

In our paper, we have chosen commonly estimated benchmarks (Cora, CiteSeer, PubMed, Amazon-Computers, and Amazon Photo) following the original papers (GRACE [77], GCA [79], and so on). Here, we also provide results and discussions about extensive benchmarks including heteophily benchmarks and large benchmarks.

**Heteophily benchmarks.** We conduct experiments on two heterophilic datasets Wikipedia-Chameleon and Wikipedia-Squirrel [36] with the GRACE method. As observed in Table 9, training with only negative samples (NO Pos) also gains benefits compared with randomly initialized models (NO Training). However, the gap between using uniformity loss (NO Pos) and using contrastive loss (Contrast) is larger than that of homophilic datasets. In conclusion, the positive-free property of GCL is more applicable to homophilic graphs. It agrees with our theoretical analysis in Section 4.2 which assumes neighbors as positive samples.

**Large benchmarks.** Here, we further consider a larger node classification benchmark OGB-arxiv [20] with 169,343 nodes and 1,166,243 edges, using the GRACE method. A node-wise similarity matrix is needed when computing the contrastive loss, but its time complexity and space usage are intolerable for large datasets. The scalability problem is one of the reasons why larger datasets are not reported in many original papers. To solve this problem, we randomly sample N=5000 nodes when computing the similarity matrix, and send the resulting matrix to the objective function. For each iteration, we repeat such sampling 5 times and use the mean loss. The random sampling strategy is simple and straightforward, and more complicated strategies will be considered in the future.

As shown in Table 10, the performance only using negative samples is on par with that using contrastive objectives. And only using positive samples on the node classification task also results in collapse. These observations are consistent with our findings.

Table 9: Test accuracy (%) on the homophily and heteophily datasets with the GRACE methods. We compare the performances of models trained the InfoNCE loss (Contrast), uniformity loss (NO Pos), alignment loss (NO Neg), and no optimization objective (NO Training). Mean accuracy with standard derivation is reported after 10 runs. Average accuracy across datasets is reported. We conduct significance testing using Wilcoxon Signed Rank Test [57], comparing the contrastive loss with other loss types. The p-value is averaged across datasets. A value below 0.05 denotes significant accuracy difference ( red ), while a value above 0.05 indicates insignificance ( green ).

| | | Homophily | | | | | Heteophily | | | |
| | | Cora | CiteSeer | PubMed | Avg | Avg p-value | Chameleon | Squirrel | Avg | Avg p-value |
|---|---|---|---|---|---|---|---|---|---|---|
| GRACE | Contrast | $84.67 \pm 1.39$ | $73.47 \pm 2.32$ | $85.80 \pm 0.16$ | 81.31 | - | $48.12 \pm 2.35$ | $33.63 \pm 1.86$ | 40.88 | - |
| | NO Training | $69.12 \pm 4.18$ | $60.60 \pm 2.59$ | $80.65 \pm 0.80$ | 70.12 | 0.0020 | $32.23 \pm 1.82$ | $25.34 \pm 1.22$ | 28.79 | 0.0020 |
| | NO Pos | $82.65 \pm 1.18$ | $73.50 \pm 2.41$ | $85.28 \pm 0.79$ | 80.48 | 0.1934 | $42.97 \pm 2.11$ | $30.48 \pm 2.25$ | 36.73 | 0.0254 |
| | NO Neg | $29.85 \pm 1.45$ | $20.42 \pm 2.26$ | $39.63 \pm 0.81$ | 29.97 | 0.0020 | $20.61 \pm 2.38$ | $19.58 \pm 1.36$ | 20.10 | 0.0020 |
| GCA | Contrast | $84.04 \pm 1.55$ | $72.63 \pm 2.68$ | $85.92 \pm 0.69$ | 80.86 | - | $46.64 \pm 2.85$ | $35.24 \pm 1.57$ | 40.94 | - |
| | NO Training | $71.25 \pm 2.32$ | $58.50 \pm 1.32$ | $80.07 \pm 0.47$ | 69.94 | 0.0020 | $33.36 \pm 2.04$ | $25.76 \pm 2.39$ | 29.56 | 0.0020 |
| | NO Pos | $83.09 \pm 2.03$ | $70.42 \pm 3.07$ | $84.68 \pm 0.63$ | 79.40 | 0.1322 | $40.17 \pm 3.93$ | $28.60 \pm 1.05$ | 34.39 | 0.0107 |
| | NO Neg | $31.40 \pm 3.61$ | $22.16 \pm 3.01$ | $39.58 \pm 0.83$ | 31.05 | 0.0020 | $21.92 \pm 4.15$ | $20.19 \pm 0.55$ | 21.10 | 0.0020 |
| ProGCL | Contrast | $85.42 \pm 3.41$ | $72.85 \pm 2.99$ | OOM | 79.14 | - | $48.38 \pm 3.65$ | $33.47 \pm 1.93$ | 40.93 | - |
| | NO Training | $79.41 \pm 0.90$ | $58.08 \pm 1.27$ | $83.54 \pm 0.83$ | 73.68 | 0.0026 | $34.21 \pm 1.15$ | $25.26 \pm 2.24$ | 29.74 | 0.0020 |
| | NO Pos | $86.76 \pm 0.52$ | $70.76 \pm 1.63$ | OOM | 78.76 | 0.2266 | $46.44 \pm 4.14$ | $30.98 \pm 4.32$ | 38.71 | 0.1064 |
| | NO Neg | $30.15 \pm 2.70$ | $21.08 \pm 1.45$ | $21.13 \pm 1.20$ | 24.12 | 0.0020 | $20.09 \pm 1.63$ | $20.46 \pm 1.57$ | 20.28 | 0.0020 |

Table 10: Test accuracy (%) on the OGB-arxiv benchmark using GRACE method with the sampled InfoNCE loss (Contrast), uniformity loss (NO Pos), and alignment loss (NO Neg).

| | Contrast | NO Pos | NO Neg |
|---|---|---|---|
| OGB-arxiv | $65.97 \pm 0.23$ | $65.49 \pm 0.32$ | $23.88 \pm 0.46$ |

## D Feature Collapse in Negative-free GCL for Node Classification

In Table 2, we find that the absence of negative samples in GCL leads to a significant performance drop for the node classification task. Numerous factors may be responsible for the suboptimal performance. Here we visualize the training process with alignment loss and InfoNCE loss to show that feature collapse is the underlying cause.

Specifically, we show the tendency of loss, average similarities of node representations $\mathbf{H} = f(\mathbf{X})$ and $\mathbf{Z} = g(\mathbf{H})$, and $L_2$ norms of weight matrices in Figure 3. From Figure 3(a), we can find that when trained with the alignment loss, the training loss steeply converges to $-1$ (optimal for the alignment loss) after the start of training. However, the similarities among node representations $\mathbf{H}$

and $\mathbf{Z}$ both unite towards one. It indicates that once the training starts, the model quickly learns the short-cut where most node representations are identical to meet the alignment loss. We also delineate $L_2$ norms of the weight matrices, which consistently converge to zero during training. As a comparison, we show the training process with InfoNCE loss in Figure 3(b). When trained with InfoNCE loss, the average similarities of node representations are relatively low and norms of weights are non-zero, showing that the collapse issue does not occur in the training process.

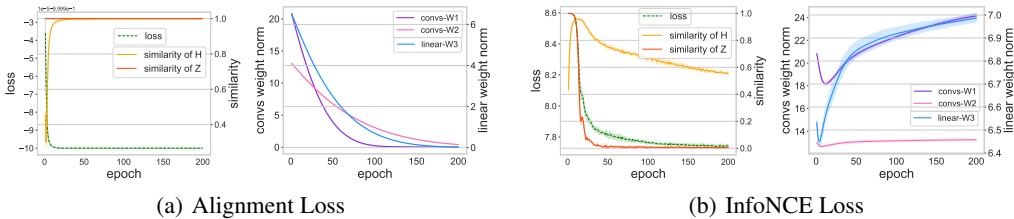

(a) Alignment Loss          (b) InfoNCE Loss

Figure 3: Tendency of loss, average similarities of node representations $\mathbf{H}$ and $\mathbf{Z}$, and $L_2$ norms of weight matrices. We choose weight matrices of the first and the second convolutional layer (Convs-W1 and Convs-W2), and the first linear layer of the projection head (Linear-W3). Experiments are conducted on Cora with GRACE.

## E  Why No-negative GCL Not Collapse in the Graph Classification

In Section 5, we observe different phenomena in the graph classification and node classification. Specifically, in the graph classification task, GCL methods achieve decent performance in the no-negative setting, while the representations collapse in the node classification task. From the architecture perspective, we find in the graph classification task, the representations learned by the projector tend to be identity, while the representations learned by the encoder escape from collapse. We suspect that learning a collapsed solution is relatively easier for the global graph representation, which can be achieved solely by the projection head.

Here, we provide some empirical insights into these conjectures. Instead of researching how to make representations not collapse in the node classification, we choose to explore *when no-negative GCL collapses in the graph classification*.

The well-known over-smoothing phenomenon states that when repeatedly applying the graph convolution, node features become indistinguishable [27]. The feature collapse is observed in contrastive learning with the alignment loss alone, where all sample features collapse to a single point. The formal equivalence established between graph convolution and the alignment loss (Theorem 4.1) reveals that the two phenomena inherently describe the same thing. To make no-negative GCL fail in graph classification, a straightforward method is stacking more layers within the encoder. Taking the MUTAG dataset as an illustrated example, we indeed find an increase in the similarities of representation $\mathbf{H}$ and $\mathbf{Z}$, and a drop in the performance (Figure 4(a)) when the layer number increases. Another choice is removing the projection head and exposing the encoder. Additionally, we increase the learning rate, whose motivation is enforcing the encoder to iterate to the collapsed solution more quickly. In Figure 4(b), we find that after removing the projection head, the encoder also collapses when the learning rate is raised to 0.01.

Besides the above two extreme cases, here we propose a more convincing method. Imitating the node-wise loss in the node classification, we transform the loss in GraphCL to an L-L version. Formally, the L-L align loss for the graph classification is:

$$\hat{\mathcal{L}}_{align} = -\frac{1}{M}\sum_{i=1}^{M}\frac{1}{N_i}\sum_{\mathbf{u}\in\mathcal{G}_i} s(\mathbf{u},\mathbf{v}), \qquad (11)$$

where $M$ denotes the number of graphs, $N_i$ denotes the number of nodes in the graph $\mathcal{G}_i$, and the positive sample $\mathbf{v}$ is the corresponding node of $\mathbf{u}$ in the augmented graph. Using this alignment loss, we train the modified GraphCL method and get a terrible test accuracy of $68.18\%$ compared to the original performance of $86.36\%$. Figure 4(c) shows that the similarities of $\mathbf{H}$ and $\mathbf{Z}$ both converge close to one during training under this loss. These observations further validate our conjecture.

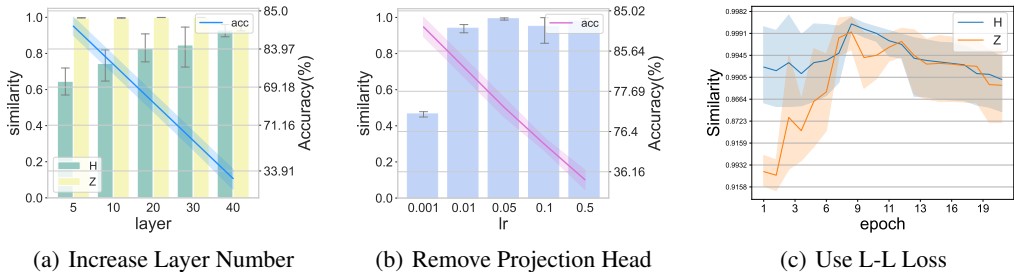

| (a) Increase Layer Number | (b) Remove Projection Head | (c) Use L-L Loss |

Figure 4: Experiments for the collapse of no-negative GCL in the graph classification. As the layer number of encoder increases, the similarity of representations **H** converges close to one and the performance degrades greatly (Figure 4(a)). A similar phenomenon is observed when removing the projection head and training the encoder with a relatively high learning rate (Figure 4(b)). Additionally, by modifying the graph-level alignment loss to a local node-wise version, we also observe a collapse in the encoder (Figure 4(c)). Experiments are conducted on MUTAG with GraphCL.

# F Proof of Theorems

## F.1 Reality of Assumption in Theorem 4.1

The augmentations in GCN's implicit alignment loss (using neighbor nodes as positive pairs) differ from those in GCL methods (like node dropping and edge perturbation). However, Table 11 demonstrates their comparable performances, suggesting the complementarity. Therefore, GCN's implicit alignment can replace positive samples under the no-positive setting, achieving good performance. This phenomenon could arise from shared domain priors: neighboring nodes have similar labels, making slight perturbations of edges/nodes inconsequential for their class membership.

Table 11: Node classification accuracy (%) under GRACE backbone with different augmentations. Mean accuracy with standard derivation is reported after 10 runs. Average accuracy across datasets is reported. We conduct significance testing using Wilcoxon Signed Rank Test [57], comparing the contrastive loss with other loss types. The p-value is averaged across datasets. A value below 0.05 denotes significant accuracy difference ( red ), while a value above 0.05 indicates insignificance ( green ).

| Augmentation | Cora | CiteSeer | PubMed | Photo | Computers | Avg. | Avg p-value |
|---|---|---|---|---|---|---|---|
| NO Training | $69.12 \pm 4.18$ | $60.60 \pm 2.59$ | $80.65 \pm 0.80$ | $68.37 \pm 3.76$ | $57.02 \pm 1.93$ | 67.15 | 0.0020 |
| GRACE augmentations | $84.67 \pm 1.39$ | $73.47 \pm 2.32$ | $85.80 \pm 0.16$ | $91.42 \pm 1.27$ | $89.01 \pm 0.60$ | 84.87 | - |
| Neighbor nodes (ours) | $84.93 \pm 2.63$ | $71.92 \pm 1.51$ | $84.72 \pm 0.32$ | $90.54 \pm 0.64$ | $86.21 \pm 0.58$ | 83.66 | 0.1668 |

## F.2 Derivation of Theorem 4.1

*Proof.* It is easy to see that under the definition of the positive samples, the alignment loss can be written equivalently as

$$\tilde{\mathcal{L}}_{\text{align}}(\mathbf{H}) = -\mathbb{E}_{x,x^+ \sim \mathcal{P}_{\mathcal{G}}(x,x^+)}[\mathbf{h}_x^\top \mathbf{h}_{x^+}] \tag{12}$$

$$= -\sum_{x,x^+} \mathcal{P}_{\mathcal{G}}(x,x^+)[\mathbf{h}_x^\top \mathbf{h}_{x^+}] \tag{13}$$

$$= -\sum_{x,x^+} [\hat{\mathbf{A}}_{x,x^+} \mathbf{h}_x^\top \mathbf{h}_{x^+}] / \sum_{x,x^+} [\hat{\mathbf{A}}_{x,x^+}] \tag{14}$$

$$= -\text{tr}\left(\mathbf{H}\hat{\mathbf{A}}\mathbf{H}^\top\right)/c, \tag{15}$$

where $c = \sum_{x,x^+} [\hat{\mathbf{A}}_{x,x^+}]$ is a constant.

Here, to maintain the feature scale, we further consider a regularization term on the norm of node features:

$$\hat{\tilde{\mathcal{L}}}_{\text{align}}(\mathbf{H}) = \tilde{\mathcal{L}}_{\text{align}}(\mathbf{H}) + \|\mathbf{H}\|^2/c. \tag{16}$$

Therefore, the gradient update of the alignment objective (Eq 6) gives the following update rule of node features $\mathbf{H}$:

$$\mathbf{H}_{\text{new}} = \mathbf{H} - \alpha\nabla_{\mathbf{H}}\hat{\tilde{\mathcal{L}}}_{\text{align}}(\mathbf{H}) \tag{17}$$
$$= \mathbf{H} - \alpha/c(-2\mathbf{AH} + 2\mathbf{H}) \tag{18}$$
$$= (1 - 2\alpha/c)\mathbf{H} + 2\alpha/c \cdot \mathbf{AH}, \tag{19}$$

where $\alpha$ is the step size. When we choose a specific learning rate $\alpha = c/2$, we recover the graph convolution operation in GCN [24]:

$$\mathbf{H}_{\text{new}} = \mathbf{AH}, \tag{20}$$

which completes the proof. □

### F.3 Derivation of Theorem 5.1

*Proof.* Denote $c = \sum_{x,x^+}[\hat{\mathbf{A}}_{x,x^+}]$ as a constant. Calculating the gradient of the uniformity loss w.r.t. each node feature $\mathbf{h}_x$ gives the following rule

$$\nabla_{\mathbf{h}_x}\tilde{\mathcal{L}}_{\text{uniform}} = 2/cP_{\mathcal{G}}(x)\sum_{x'}\mathbf{A}_{x,x'}\mathbf{h}_{x'}. \tag{21}$$

In a matrix form, we have

$$\nabla_{\mathbf{H}}\tilde{\mathcal{L}}_{\text{uniform}} = 2/c\mathbf{DAH}, \tag{22}$$

where $\mathbf{D}$ is the diagonal matrix containing $\mathcal{P}(x) = \sum_{x'}\mathbf{A}_{x,x'}, \forall x \in \mathcal{V}$.

Therefore, the gradient descent update of the defined uniformity loss gives

$$\mathbf{H}_{\text{new}} = \mathbf{H} - \alpha\nabla_{\mathbf{H}}\tilde{\mathcal{L}}_{\text{uniform}} = \mathbf{H} - 2/c\mathbf{DAH}, \tag{23}$$

where $\alpha$ is the step size. It is easy to see its equivalence to the ContraNorm update. □

### F.4 Derivation of Theorem 5.2

*Proof.* Combining Theorem 4.1 and Theorem 5.1, we can directly obtain Theorem 5.2 as a corollary.
□

## G Discussion on More GCL Methods

The contrastive mode has three mainstreams: local-to-local (L-L), global-to-global (G-G), and global-to-local (G-L) [78]. For the local-to-local perspective, the corresponding nodes in the two augmented views of a graph are seen as positive pairs while all the other node pairs are negative ones. Global-to-global mode is often used when there are multiple graphs, and contrastive objects are the global representations of augmented views. In this mode, augmented views of the same graph are positives and all the other graph pairs are negatives. For the global-to-local perspective, positive pairs are taken as the global representation and nodes of augmented views for the corresponding graph, and negative pairs are the global representation and nodes of augmented views for other graphs.

In previous sections, we investigate the GCL methods with L-L or G-G modes, and the G-L mode on the graph classification (like InfoGraph). In this section, we discuss two methods of the G-L mode on node classification task: DGI [51] and MVGRL [14]. For experiments, we use the same settings as in Section 4. As seen from Table 12, there is an obvious degeneration in accuracy when no positive samples or negative samples are used, which is close to the no training setting. Recall that we find the positive samples are not needed in Section 4, and the observations on DGI and MVGRL seem to contradict our arguments. Here we attribute the inconsistency to the flaw in the methods themselves.

We start with an intriguing finding on DGI. Here we disorder the contrastive correspondence with a wrong view as global representations. Specifically, we take the local representation of the graph and its global representation as *negatives*, while local representations and global representations of the

corrupted view are seen as *positives*. Note that the corruption operation in DGI is used to generate negative samples by shuffling rows of node attributes. See Figure 5 for illustration. We compare the disordered version with the original DGI in Table 13, and find using a wrong view as global representations does not affect performance. It implies that global representations lose efficacy in this framework. Inspired by Zheng et al. [76], we compare the two global representations and find they are nearly identical with every dimension being about 0.5. Extensive experiments also show the global representation is a constant vector for inappropriate usage of the Sigmoid function in both DGI and MVGRL [76].

This finding explains why the loss without positive samples does not work. Trained with such loss, node representations are only enforced to be far away from a constant vector, which gives no semantic guarantee. However, after adding positive samples to loss, the model learns to pull positive samples near a constant vector, while pushing negative samples away from such vector. It intrinsically achieves the goal of contrastive learning by gathering positives and repulsing negatives simultaneously. Thus the model trained with both positive and negative samples can obtain satisfying performance, explaining why DGI works with constant global representations.

Table 12: Test accuracy (%) of node classification benchmarks using DGI and MVGRL methods. We compare the performances of models trained with JSD loss (Contrast), loss part only involving negative pairs (NO Pos), loss only involving positive pairs (NO Neg), and no optimization objective (NO Training). Mean accuracy with standard derivation is reported after 10 runs. We conduct significance testing using Wilcoxon Signed Rank Test [57], comparing the contrastive loss with other loss types. The p-value is averaged across datasets. A value below 0.05 denotes significant accuracy difference ( red ), while a value above 0.05 indicates insignificance ( green ).

| Method | Loss | Cora | CiteSeer | PubMed | Photo | Computers | Chameleon | Squirrel | Avg | Avg p-value |
|---|---|---|---|---|---|---|---|---|---|---|
| DGI | Contrast | $83.38 \pm 2.67$ | $72.07 \pm 2.37$ | $84.77 \pm 0.71$ | $88.10 \pm 1.81$ | $83.35 \pm 0.71$ | $39.56 \pm 2.86$ | $34.55 \pm 0.88$ | 69.40 | - |
| | NO Training | $69.78 \pm 3.39$ | $55.15 \pm 2.09$ | $79.56 \pm 1.35$ | $69.08 \pm 3.30$ | $56.03 \pm 1.97$ | $31.44 \pm 1.70$ | $24.57 \pm 1.22$ | 55.09 | 0.0020 |
| | NO Pos | $66.84 \pm 3.54$ | $54.79 \pm 3.33$ | $78.25 \pm 0.99$ | $58.34 \pm 3.92$ | $71.98 \pm 1.38$ | $35.81 \pm 2.34$ | $26.99 \pm 0.20$ | 56.14 | 0.0020 |
| | NO Neg | $67.35 \pm 4.61$ | $58.17 \pm 2.57$ | $77.23 \pm 1.05$ | $62.75 \pm 3.75$ | $72.66 \pm 1.48$ | $31.62 \pm 4.06$ | $27.75 \pm 1.85$ | 56.79 | 0.0022 |
| MVGRL | Contrast | $84.41 \pm 1.44$ | $75.27 \pm 0.79$ | $85.62 \pm 0.63$ | $89.23 \pm 1.52$ | $79.58 \pm 0.15$ | $42.45 \pm 2.43$ | $33.97 \pm 2.54$ | 70.08 | - |
| | NO Training | $77.94 \pm 2.23$ | $58.92 \pm 2.88$ | $82.13 \pm 0.63$ | $81.15 \pm 3.25$ | $69.07 \pm 0.40$ | $32.23 \pm 1.94$ | $24.41 \pm 1.10$ | 60.84 | 0.0022 |
| | NO Pos | $75.44 \pm 1.42$ | $61.08 \pm 2.48$ | $81.26 \pm 1.30$ | $36.03 \pm 1.57$ | $38.36 \pm 0.55$ | $36.86 \pm 2.56$ | $29.98 \pm 1.52$ | 51.29 | 0.0020 |
| | NO Neg | $54.93 \pm 4.67$ | $35.03 \pm 5.20$ | $56.26 \pm 1.91$ | $36.47 \pm 2.37$ | $38.36 \pm 0.56$ | $29.34 \pm 2.04$ | $28.06 \pm 1.66$ | 39.78 | 0.0020 |

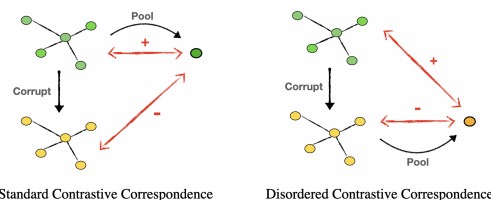

Standard Contrastive Correspondence     Disordered Contrastive Correspondence

Figure 5: Illustration for disordering contrastive correspondence of views on DGI.

Table 13: Test accuracy (%) of DGI in standard contrastive correspondence (Std) and disordered correspondence (Dis).

| Method | Contrast | Cora | CiteSeer | PubMed |
|---|---|---|---|---|
| DGI | Std. | $83.38 \pm 2.68$ | $72.07 \pm 2.37$ | $84.77 \pm 0.71$ |
| | Dis. | $83.35 \pm 2.68$ | $72.04 \pm 2.17$ | $84.70 \pm 0.68$ |

## H   Extensive Experiments of ContraNorm.

**ContraNorm in different GCL methods.** In Table 5, we show that by simply incorporating the normalization layer into the encoder, the collapse issue can be rooted out for the GRACE method. In this section, we incorporate ContraNorm into multiple GCL methods under the no-negative setting.

The results are shown in Table 14. It is obvious that for these GCL methods, applying ContraNorm when there are no negative samples achieves comparable performance with models trained with the contrastive loss (both positive and negative samples). The extensive experiments validate the effectiveness of ContraNorm across different GCL methods.

**Combining ContraNorm with MLP.** In Table 15, we show that ContraNorm could also boost MLP performance significantly (59.49%→73.19%) under the "No Neg" loss, and attain similar performance to MLP trained with contrastive loss (77.07%), which also aligns well with our theory and empirical observations on GCN.

**ContraNorm under SimCLR backbone.** In Table 16, we conduct experiments replacing uniformity loss with ContraNorm under the SimLCR backbone. It is shown that ContraNorm can not replace the uniformity loss in SimCLR. We conjecture it is a graph-specific technique and leave the analysis for the future work.

Table 14: Node classification accuracy (%) with GCN / GCN+ContraNorm using GCL methods. Mean accuracy with standard derivation is reported after 10 runs. Average accuracy across datasets is reported. We conduct significance testing using Wilcoxon Signed Rank Test [57], comparing the default setting (first line) with others. The p-value is averaged across datasets. A value below 0.05 denotes significant accuracy difference ( red ), while a value above 0.05 indicates insignificance ( green ). OOM denotes out of memory.

| Method | Loss | Encoder | Cora | CiteSeer | PubMed | Photo | Computers | Avg | Avg p-value |
|---|---|---|---|---|---|---|---|---|---|
| | Contrast | GCN | $84.67 \pm 1.39$ | $73.47 \pm 2.32$ | $85.80 \pm 0.16$ | $91.42 \pm 1.27$ | $89.01 \pm 0.60$ | 84.87 | - |
| GRACE | NO Neg | GCN | $29.85 \pm 1.45$ | $20.42 \pm 2.26$ | $39.63 \pm 0.81$ | $25.10 \pm 1.74$ | $36.84 \pm 1.30$ | 30.37 | 0.0020 |
| | NO Neg | GCN + CN | $82.35 \pm 2.28$ | $72.25 \pm 1.86$ | $83.30 \pm 0.63$ | $92.43 \pm 0.82$ | $84.48 \pm 1.01$ | 82.96 | 0.1520 |
| | Contrast | GCN | $84.04 \pm 1.55$ | $72.63 \pm 2.68$ | $85.92 \pm 0.69$ | $93.07 \pm 0.66$ | $86.58 \pm 0.75$ | 84.45 | - |
| GCA | NO Neg | GCN | $31.40 \pm 3.61$ | $22.16 \pm 3.01$ | $39.58 \pm 0.83$ | $28.13 \pm 1.14$ | $37.34 \pm 0.95$ | 31.72 | 0.0020 |
| | NO Neg | GCN + CN | $82.21 \pm 1.29$ | $72.87 \pm 0.98$ | $82.40 \pm 0.78$ | $92.47 \pm 0.96$ | $86.15 \pm 0.58$ | 83.22 | 0.2125 |
| | Contrast | GCN | $85.42 \pm 3.41$ | $72.85 \pm 2.99$ | OOM | $93.81 \pm 0.48$ | $86.35 \pm 1.28$ | 84.61 | - |
| ProGCL | NO Neg | GCN | $30.15 \pm 2.70$ | $21.08 \pm 1.45$ | $21.13 \pm 1.20$ | $4.88 \pm 0.33$ | $3.11 \pm 0.65$ | 16.07 | 0.0020 |
| | NO Neg | GCN + CN | $80.00 \pm 1.75$ | $73.35 \pm 1.17$ | $84.02 \pm 0.91$ | $93.59 \pm 0.38$ | $85.67 \pm 0.43$ | 83.33 | 0.2336 |

Table 15: Node classification accuracy (%) with MLP / MLP+ContraNorm using GRACE. Mean accuracy with standard derivation is reported after 10 runs. Average accuracy across datasets is reported. We conduct significance testing using Wilcoxon Signed Rank Test [57], comparing the default setting (first line) with others. The p-value is averaged across datasets. A value below 0.05 denotes significant accuracy difference ( red ), while a value above 0.05 indicates insignificance ( green ). OOM denotes out of memory.

| Loss | Encoder | Cora | CiteSeer | PubMed | Photo | Computers | Avg. | Avg p-value. |
|---|---|---|---|---|---|---|---|---|
| Contrast | MLP | $67.72 \pm 0.88$ | $65.51 \pm 2.63$ | $83.29 \pm 0.49$ | $87.92 \pm 0.59$ | $80.89 \pm 1.21$ | 77.07 | - |
| NO Training | MLP | $40.66 \pm 2.49$ | $42.81 \pm 4.82$ | $78.53 \pm 0.90$ | $62.12 \pm 0.97$ | $57.97 \pm 1.13$ | 56.42 | 0.0020 |
| NO Neg | MLP | $51.69 \pm 3.04$ | $50.36 \pm 1.14$ | $79.00 \pm 0.63$ | $61.33 \pm 0.75$ | $55.07 \pm 0.99$ | 59.49 | 0.0020 |
| NO Neg | MLP + CN | $62.87 \pm 0.84$ | $62.16 \pm 3.11$ | $81.51 \pm 0.60$ | $83.03 \pm 1.59$ | $76.40 \pm 1.09$ | 73.19 | 0.0660 |

Table 16: Image classification accuracy (%) under SimCLR backbone on CIFAR10.

| Loss & Encoder | SimCLR & ResNet | Uniform & ResNet | Align & ResNet | Align & ResNet + CN |
|---|---|---|---|---|
| Test Acc (%) | 82.4 | 20.3 | 18.6 | 20.9 |

# I  Extensive experiments for Gaussian Augmentations

**Gaussian Augmentations under Different Loss Settings.** In Section 6, we perform experiments using the GRACE method with different augmentations under the InfoNCE loss. Here, we further

report results under different losses in Table 17. For loss without negative samples, the average performance gap between domain-specific augmentations and noise augmentations is only $0.74\%$. When no augmentations, the performance drops $4.88\%$. We conjecture that when no negative samples exist, the application of augmentations brings diversity in representations, thus making collapse more difficult. For contrastive loss and loss without positive samples, the gap between domain-specific augmentations and noise augmentations is also narrow.

**Gaussian Augmentations Using MLP As the Encoder.** We compare augmentations with MLP backbone. In Table 18, while GCN exhibits similar performance with different augmentations, FM+EP notably surpasses Gaussian noise for MLP. This also correlates with GCN's implicit alignment mechanism (Theorem 4.1). Additional augmentations minimally affect GCN due to the existing alignment mechanism. However, for MLP without this implicit bias, graph-specific augmentations like FM+EP remain informative for learning proper graph invariance.

Table 17: Test accuracy (%) of node classification benchmarks using GRACE method with different augmentations under three loss settings. We compare no augmentations (NO Aug), domain-agnostic augmentations (Gaussian), and default domain-specific augmentations (FM+EP). Average accuracy and p-value are reported. We conduct significance testing using Wilcoxon Signed Rank Test [57], comparing the default augmentation with other settings. The p-value is averaged across datasets. A value below 0.05 denotes significant accuracy difference ( red ), while a value above 0.05 indicates insignificance ( green ).

| Loss | Encoder | Aug | Cora | CiteSeer | PubMed | Photo | Computers | Avg | Avg p-value |
|------|---------|-----|------|----------|--------|-------|-----------|-----|-------------|
| Contrast | GCN | FM+EP | 84.67 ± 1.39 | 73.47 ± 2.32 | 85.80 ± 0.16 | 91.42 ± 1.27 | 89.01 ± 0.60 | 84.87 | - |
| | | Gaussian | 82.72 ± 2.38 | 72.60 ± 1.21 | 85.24 ± 0.61 | 91.32 ± 1.37 | 82.77 ± 1.09 | 82.93 | 0.1816 |
| | | NO Aug | 79.56 ± 2.18 | 71.83 ± 1.83 | 84.68 ± 0.58 | 90.99 ± 1.26 | 82.83 ± 0.86 | 81.98 | 0.1008 |
| NO Pos | GCN | FM+EP | 82.65 ± 1.18 | 73.50 ± 2.41 | 85.28 ± 0.79 | 91.32 ± 0.10 | 84.40 ± 0.43 | 83.43 | - |
| | | Gaussian | 80.04 ± 1.93 | 70.84 ± 1.85 | 84.88 ± 0.89 | 91.33 ± 1.18 | 83.26 ± 1.24 | 82.07 | 0.1840 |
| | | NO Aug | 79.37 ± 2.30 | 71.80 ± 1.84 | 84.69 ± 0.63 | 90.92 ± 1.21 | 82.49 ± 0.87 | 81.85 | 0.1176 |
| NO Neg | GCN + CN | FM+EP | 82.35 ± 2.28 | 72.25 ± 1.86 | 83.30 ± 0.63 | 92.43 ± 0.82 | 84.48 ± 1.01 | 82.96 | - |
| | | Gaussian | 79.08 ± 2.47 | 72.43 ± 1.32 | 83.55 ± 0.22 | 91.59 ± 1.19 | 84.48 ± 1.07 | 82.23 | 0.2750 |
| | | NO Aug | 75.59 ± 3.45 | 66.98 ± 3.40 | 82.14 ± 1.28 | 81.91 ± 1.42 | 83.79 ± 1.14 | 78.08 | 0.0688 |

Table 18: Node classification accuracy (%) under GRACE backbone with MLP using different augmentations.

| Augmentation | Encoder | Cora | CiteSeer | PubMed | Photo | Computers | Avg. |
|--------------|---------|------|----------|--------|-------|-----------|------|
| NO Aug | MLP | 58.09 ± 2.96 | 62.69 ± 1.21 | 80.62 ± 1.01 | 84.42 ± 0.64 | 73.30 ± 1.10 | 71.82 |
| FM + EP | MLP | 67.72 ± 0.88 | 65.51 ± 2.63 | 83.29 ± 0.49 | 87.92 ± 0.59 | 80.89 ± 1.21 | 77.07 |
| Gaussian noise | MLP | 61.47 ± 3.36 | 63.23 ± 2.41 | 81.90 ± 0.57 | 84.03 ± 1.07 | 75.47 ± 0.70 | 73.22 |

Table 19: Node classification accuracy (%) under GRACE backbone with no-training.

| Loss | Encoder | Cora | CiteSeer | PubMed | Photo | Computers | Avg. |
|------|---------|------|----------|--------|-------|-----------|------|
| Contrast | GCN | 84.67 ± 1.39 | 73.47 ± 2.32 | 85.80 ± 0.16 | 91.42 ± 1.27 | 89.01 ± 0.60 | 84.87 |
| NO Training | GCN | 69.12 ± 4.18 | 60.60 ± 2.59 | 80.65 ± 0.80 | 68.37 ± 3.76 | 57.02 ± 1.93 | 67.15 |
| NO Training | GCN+CN | 69.63 ± 4.08 | 59.82 ± 2.73 | 81.73 ± 0.97 | 91.37 ± 0.60 | 85.64 ± 0.70 | 77.64 |

## J   Can GCL trained without both positive and negative pairs?

In Section 4, and in Section 5. A natural question arises: can GRACE with GCN and ContraNorm be trained without positive AND negative pairs? Removing both positive and negative samples renders the InfoNCE loss empty, actually corresponding to the "No Training" baseline which typically performs much worse. Results in Table 19 further show the performance remains poor even when adding

ContraNorm to GCN. This is because despite the implicit alignment and uniformity mechanisms, without any training objective, the model parameters in GCN+CN cannot be properly trained to fit the dataset.

# K    Results of the Fine-tuning Protocol

In this section, we provide the fine-tuning protocol results of main experiments of our paper. Specifically, we add a linear classification head after the encoder. In the fine-tuning phase, we fine-tune the whole networks according to downstream tasks, with the learning rate selected from $[0.01, 0.001]$ and the number of epochs selected from $[100, 200, 500]$. In Table 20(a) and Table 20(b), we report the fine-tuning results for the node classification task with GRACE and DGI methods, and for the graph classification tasks with GraphCL method, respectively. Sharing the same conclusion as the linear probing protocol, only using negative samples achieves comparable performance as that using contrastive objectives. On the other hand, for the node classification task, only using positive samples escapes severe collapse. We think the guidance of true labels in the fine-tuning helps the networks relearn parameters and thus prevents collapse.

Furthermore, we report the fine-tuning results about augmentations in Table 20(c). For the default augmentations (FM+PE), we set the ratio of each augmentation to 0.2 to save engineering effort. For a fair comparison, the standard deviation $\sigma$ of the random Gaussian noise is fixed to 1e-4. Other hyperparameters are the same across the three augmentation settings (FM+PE, Gaussian, and NO Aug). As seen from the table, in the fine-tuning evaluation setting, random noise augmentation is on average the best for each loss type. It further justifies our analysis that domain-agnostic augmentations are enough for GCL.

Table 20: Fine-tuning accuracy (%) using GCL methods. Mean accuracy with standard derivation is reported after 10 runs. Average accuracy across datasets is reported. We conduct significance testing using Wilcoxon Signed Rank Test [57], comparing the contrastive loss and other loss types. The p-value is averaged across datasets. A value below 0.05 denotes significant accuracy difference ( red ), while a value above 0.05 denotes insignificance ( green ).

(a) Fine-tuning accuracy (%) of node classification benchmarks using GCL methods.

| Method | Loss | Cora | CiteSeer | PubMed | Photo | Computers | Chameleon | Squirrel | Avg | Avg p-value |
|---|---|---|---|---|---|---|---|---|---|---|
| GRACE | Contrast | $85.15 \pm 3.07$ | $74.19 \pm 3.66$ | $84.64 \pm 2.47$ | $92.89 \pm 0.56$ | $88.92 \pm 1.27$ | $39.56 \pm 3.76$ | $33.13 \pm 4.33$ | 71.21 | - |
| | NO Pos | $84.49 \pm 3.50$ | $74.07 \pm 3.92$ | $82.38 \pm 2.36$ | $92.84 \pm 0.51$ | $89.45 \pm 1.14$ | $38.60 \pm 3.99$ | $31.40 \pm 3.76$ | 70.46 | 0.3139 |
| | NO Neg | $81.62 \pm 4.05$ | $69.52 \pm 4.46$ | $83.87 \pm 2.65$ | $92.05 \pm 0.89$ | $89.07 \pm 1.03$ | $35.98 \pm 5.13$ | $30.48 \pm 2.54$ | 68.94 | 0.1398 |
| DGI | Contrast | $85.66 \pm 2.39$ | $74.55 \pm 1.68$ | $85.69 \pm 0.26$ | $92.94 \pm 0.88$ | $90.03 \pm 0.79$ | $42.01 \pm 6.07$ | $32.74 \pm 5.49$ | 71.95 | - |
| | NO Pos | $86.91 \pm 2.16$ | $74.79 \pm 0.92$ | $85.49 \pm 0.25$ | $92.73 \pm 0.69$ | $89.45 \pm 0.65$ | $42.01 \pm 6.05$ | $31.98 \pm 5.73$ | 71.91 | 0.4395 |
| | NO Neg | $85.00 \pm 2.39$ | $74.97 \pm 1.08$ | $85.44 \pm 0.38$ | $92.73 \pm 0.67$ | $90.13 \pm 0.66$ | $42.97 \pm 6.52$ | $32.13 \pm 5.24$ | 71.91 | 0.3672 |

(b) Fine-tuning accuracy (%) of graph classification benchmarks using GCL methods.

| Method | Loss | MUTAG | PTC-MR | PROTEINS | IMDB-BINARY | IMDB-MULTI | REDDIT-BINARY | Avg | Avg p-value |
|---|---|---|---|---|---|---|---|---|---|
| GraphCL | Contrast | $93.48 \pm 2.52$ | $80.64 \pm 4.09$ | $79.88 \pm 0.43$ | $64.53 \pm 1.32$ | $43.64 \pm 0.63$ | $79.67 \pm 1.82$ | 73.64 | - |
| | NO Pos | $93.12 \pm 0.74$ | $80.45 \pm 4.85$ | $79.34 \pm 3.10$ | $63.13 \pm 1.55$ | $42.24 \pm 0.31$ | $76.73 \pm 4.23$ | 72.50 | 0.1966 |
| | NO Neg | $93.11 \pm 1.14$ | $80.36 \pm 3.64$ | $79.01 \pm 3.69$ | $62.37 \pm 2.81$ | $41.60 \pm 0.47$ | $76.75 \pm 2.90$ | 72.20 | 0.1400 |

(c) Fine-tuning accuracy (%) of node classification benchmarks using GRACE method with different augmentations under three loss settings.

| Loss | Aug | Cora | CiteSeer | PubMed | Photo | Computers | Chameleon | Squirrel | Avg | Avg p-value |
|---|---|---|---|---|---|---|---|---|---|---|
| Contrast | FM+EP | $85.15 \pm 3.07$ | $74.19 \pm 3.66$ | $84.64 \pm 2.47$ | $92.89 \pm 0.56$ | $88.92 \pm 1.27$ | $39.56 \pm 3.76$ | $33.13 \pm 4.33$ | 71.21 | - |
| | Gaussian | $86.69 \pm 2.39$ | $74.91 \pm 2.98$ | $84.52 \pm 2.05$ | $92.94 \pm 1.02$ | $88.94 \pm 1.14$ | $42.62 \pm 6.55$ | $31.55 \pm 4.64$ | 71.74 | 0.2829 |
| | NO Aug | $85.00 \pm 3.20$ | $74.07 \pm 3.92$ | $82.64 \pm 2.69$ | $93.10 \pm 0.42$ | $89.58 \pm 1.20$ | $37.03 \pm 4.28$ | $31.59 \pm 4.49$ | 70.43 | 0.2531 |
| NO Pos | FM+EP | $84.49 \pm 3.50$ | $74.07 \pm 3.92$ | $82.38 \pm 2.36$ | $92.84 \pm 0.51$ | $89.45 \pm 1.14$ | $38.60 \pm 3.99$ | $31.40 \pm 3.76$ | 70.46 | - |
| | Gaussian | $86.40 \pm 2.84$ | $74.67 \pm 3.90$ | $83.95 \pm 1.72$ | $92.73 \pm 1.52$ | $88.72 \pm 1.18$ | $40.79 \pm 6.03$ | $30.17 \pm 4.73$ | 71.06 | 0.2609 |
| | NO Aug | $85.00 \pm 3.20$ | $74.07 \pm 3.92$ | $82.42 \pm 2.57$ | $92.97 \pm 0.58$ | $89.49 \pm 1.10$ | $38.43 \pm 3.88$ | $31.59 \pm 4.48$ | 70.57 | 0.3859 |
| NO Neg | FM+EP | $81.62 \pm 4.05$ | $69.52 \pm 4.46$ | $83.87 \pm 2.65$ | $92.05 \pm 0.89$ | $89.07 \pm 1.03$ | $35.98 \pm 5.13$ | $30.48 \pm 2.54$ | 68.94 | - |
| | Gaussian | $84.26 \pm 2.80$ | $72.46 \pm 4.75$ | $84.49 \pm 1.97$ | $91.56 \pm 1.75$ | $88.37 \pm 1.73$ | $38.25 \pm 2.54$ | $28.25 \pm 2.81$ | 69.66 | 0.3273 |
| | NO Aug | $80.96 \pm 5.24$ | $71.38 \pm 5.59$ | $82.45 \pm 2.69$ | $92.03 \pm 2.12$ | $86.16 \pm 5.95$ | $33.97 \pm 4.41$ | $26.76 \pm 2.49$ | 67.67 | 0.2854 |

# L    Extended Evaluation for GCL and VCL Under the Same Splitting

Across our experiments, we follow the standard setting and data splits in each domain. To further resolve concerns on the consistency of evaluation setups, we unify the evaluation settings for GCL and VCL: we all train a linear classifier with an Adam optimizer and randomly split the dataset with the same train-test ratio as 9:1. The results are shown in the following table.

As can be seen in Table 21 , the observations are consistent with the findings in our paper. GCL and VCL present apparent differences in the properties of no-positives, no-negatives and random Gaussian noise augmentations. The supervised information ratio shows little effect on the conclusions. For the node classification task, evaluations are under data splits of 1:9 in the paper and 9:1 here, but the findings are kept unchanged.

Table 21: Evaluation results of GCL and VCL under the same data split setting. Average accuracy and p-value are reported. We conduct significance testing using Wilcoxon Signed Rank Test [57], comparing the default augmentation with other settings. The p-value is averaged across datasets. A value below 0.05 denotes a significant accuracy difference ( red ), while a value above 0.05 indicates insignificance ( green ).

(a) Image classification test accuracy (%) with SimCLR backbone on CIFAR10

| Loss | Augmentation | Avg. |
|---|---|---|
| NO Training | Default Aug | 22.5 |
| InfoNCE | Default Aug | 82.4 |
| Alignment | Default Aug | 18.6 |
| Uniformity | Default Aug | 20.3 |
| InfoNCE | Gaussian | 38.6 |

(b) Node classification test accuracy (%) with GRACE as backbone

| Loss | Encoder | Cora | CiteSeer | PubMed | Photo | Computers | Avg. | Avg p-value. |
|---|---|---|---|---|---|---|---|---|
| Contrast | GCN | $86.27 \pm 1.31$ | $75.14 \pm 2.11$ | $86.02 \pm 0.87$ | $92.10 \pm 0.97$ | $83.01 \pm 0.66$ | 84.51 | - |
| NO Training | GCN | $76.31 \pm 2.69$ | $68.23 \pm 2.31$ | $83.22 \pm 0.90$ | $69.20 \pm 1.06$ | $56.95 \pm 1.36$ | 70.78 | 0.0020 |
| NO Pos | GCN | $86.35 \pm 1.68$ | $74.05 \pm 1.34$ | $85.50 \pm 0.62$ | $91.42 \pm 0.98$ | $83.33 \pm 0.99$ | 84.13 | 0.3273 |
| NO Neg | GCN | $29.74 \pm 1.92$ | $21.62 \pm 1.89$ | $40.30 \pm 0.41$ | $26.12 \pm 1.88$ | $38.10 \pm 1.35$ | 31.18 | 0.0020 |
| NO Neg | GCN + CN | $88.49 \pm 1.94$ | $77.12 \pm 1.18$ | $85.16 \pm 0.84$ | $94.67 \pm 0.70$ | $85.61 \pm 1.34$ | 86.21 | 0.1324 |

(c) Graph classification test accuracy (%) with GraphCL as backbone

| Loss | MUTAG | PTC_MR | PROTEINS | IMDB-B | IMDB-M | REDDIT-B | Avg. | Avg p-value. |
|---|---|---|---|---|---|---|---|---|
| Contrast | $91.58 \pm 2.58$ | $71.43 \pm 6.26$ | $78.21 \pm 3.73$ | $76.40 \pm 1.50$ | $52.93 \pm 4.12$ | $88.70 \pm 1.03$ | 76.54 | - |
| NO Training | $86.32 \pm 5.37$ | $65.71 \pm 3.13$ | $72.14 \pm 4.50$ | $70.80 \pm 2.14$ | $44.13 \pm 1.65$ | $75.30 \pm 2.01$ | 69.07 | 0.0104 |
| NO Pos | $92.63 \pm 6.32$ | $73.71 \pm 3.33$ | $77.32 \pm 4.02$ | $76.60 \pm 1.85$ | $52.00 \pm 3.86$ | $86.90 \pm 1.39$ | 76.53 | 0.6086 |
| NO Neg | $91.58 \pm 8.55$ | $69.71 \pm 8.40$ | $78.75 \pm 3.21$ | $77.80 \pm 1.94$ | $52.80 \pm 3.80$ | $87.00 \pm 0.84$ | 76.23 | 0.3910 |

(d) Node classification test accuracy (%) with GRACE as backbone with different augmentations

| Augmentation | Cora | CiteSeer | PubMed | Photo | Computers | Avg. | Avg p-value. |
|---|---|---|---|---|---|---|---|
| NO Aug | $85.76 \pm 1.72$ | $74.77 \pm 1.02$ | $85.65 \pm 0.88$ | $90.14 \pm 1.40$ | $82.38 \pm 1.00$ | 83.74 | 0.2227 |
| FM+EP | $86.27 \pm 1.31$ | $75.14 \pm 2.11$ | $86.02 \pm 0.87$ | $92.10 \pm 0.97$ | $83.01 \pm 0.66$ | 84.51 | - |
| Gaussian | $86.35 \pm 1.82$ | $74.35 \pm 1.90$ | $86.09 \pm 0.58$ | $90.72 \pm 1.81$ | $83.17 \pm 0.81$ | 84.14 | 0.3848 |

