| 84.67 ± 1.39 | 73.47 ± 2.32 | 85.80 ± 0.16 | 81.31 | - | 48.12 ± 2.35 | 33.63 ± 1.86 | 40.88 | - |
| | NO Training | 69.12 ± 4.18 | 60.60 ± 2.59 | 80.65 ± 0.80 | 70.12 | 0.0020 | 32.23 ± 1.82 | 25.34 ± 1.22 | 28.79 | 0.0020 |
| | NO Pos | 82.65 ± 1.18 | 73.50 ± 2.41 | 85.28 ± 0.79 | 80.48 | 0.1934 | 42.97 ± 2.11 | 30.48 ± 2.25 | 36.73 | 0.0254 |
| | NO Neg | 29.85 ± 1.45 | 20.42 ± 2.26 | 39.63 ± 0.81 | 29.97 | 0.0020 | 20.61 ± 2.38 | 19.58 ± 1.36 | 20.10 | 0.0020 |
| GCA | Contrast | 84.04 ± 1.55 | 72.63 ± 2.68 | 85.92 ± 0.69 | 80.86 | - | 46.64 ± 2.85 | 35.24 ± 1.57 | 40.94 | - |
| | NO Training | 71.25 ± 2.32 | 58.50 ± 1.32 | 80.07 ± 0.47 | 69.94 | 0.0020 | 33.36 ± 2.04 | 25.76 ± 2.39 | 29.56 | 0.0020 |
| | NO Pos | 83.09 ± 2.03 | 70.42 ± 3.07 | 84.68 ± 0.63 | 79.40 | 0.1322 | 40.17 ± 3.93 | 28.60 ± 1.05 | 34.39 | 0.0107 |
| | NO Neg | 31.40 ± 3.61 | 22.16 ± 3.01 | 39.58 ± 0.83 | 31.05 | 0.0020 | 21.92 ± 4.15 | 20.19 ± 0.55 | 21.10 | 0.0020 |
| ProGCL | Contrast | 85.42 ± 3.41 | 72.85 ± 2.99 | OOM | 79.14 | - | 48.38 ± 3.65 | 33.47 ± 1.93 | 40.93 | - |
| | NO Training | 79.41 ± 0.90 | 58.08 ± 1.27 | 83.54 ± 0.83 | 73.68 | 0.0026 | 34.21 ± 1.15 | 25.26 ± 2.24 | 29.74 | 0.0020 |
| | NO Pos | 86.76 ± 0.52 | 70.76 ± 1.63 | OOM | 78.76 | 0.2266 | 46.44 ± 4.14 | 30.98 ± 4.32 | 38.71 | 0.1064 |
| | NO Neg | 30.15 ± 2.70 | 21.08 ± 1.45 | 21.13 ± 1.20 | 24.12 | 0.0020 | 20.09 ± 1.63 | 20.46 ± 1.57 | 20.28 | 0.0020 |

**Large benchmarks.** Here, we further consider a larger node classification benchmark OGB-arxiv [18] with 169,343 nodes and 1,166,243 edges, using the GRACE method. A node-wise similarity matrix is needed when computing the contrastive loss, but its time complexity and space usage are intolerable for large datasets. The scalability problem is one of the reasons why larger datasets are not reported in many original papers. To solve this problem, we randomly sample N=5000 nodes when computing the similarity matrix, and send the resulting matrix to the objective function. For each iteration, we repeat such sampling 5 times and use the mean loss. The random sampling strategy is simple and straightforward, and more complicated strategies will be considered in the future.

As shown in Table 10, the performance only using negative samples is on par with that using contrastive objectives. And only using positive samples on the node classification task also results in collapse. These observations are consistent with our findings.

Table 10: Test accuracy (%) on the OGB-arxiv benchmark using GRACE method with the sampled InfoNCE loss (Contrast), uniformity loss (NO Pos), and alignment loss (NO Neg).

|  | Contrast | NO Pos | NO Neg |
|---|---|---|---|
| OGB-arxiv | $65.97 \pm 0.23$ | $65.49 \pm 0.32$ | $23.88 \pm 0.46$ |

## D    Feature Collapse in Negative-free GCL for Node Classification

In Table 2, we find that the absence of negative samples in GCL leads to a significant performance drop for the node classification task. Numerous factors may be responsible for the suboptimal performance. Here we visualize the training process with alignment loss and InfoNCE loss to show that feature collapse is the underlying cause.

Specifically, we show the tendency of loss, average similarities of node representations $\mathbf{H} = f(\mathbf{X})$ and $\mathbf{Z} = g(\mathbf{H})$, and $L_2$ norms of weight matrices in Figure 3. From Figure 3(a), we can find that when trained with the alignment loss, the training loss steeply converges to $-1$ (optimal for the alignment loss) after the start of training. However, the similarities among node representations $\mathbf{H}$ and $\mathbf{Z}$ both unite towards one. It indicates that once the training starts, the model quickly learns the short-cut where most node representations are identical to meet the alignment loss. We also delineate $L_2$ norms of the weight matrices, which consistently converge to zero during training. As a comparison, we show the training process with InfoNCE loss in Figure 3(b). When trained with InfoNCE loss, the average similarities of node representations are relatively low and norms of weights are non-zero, showing that the collapse issue does not occur in the training process.

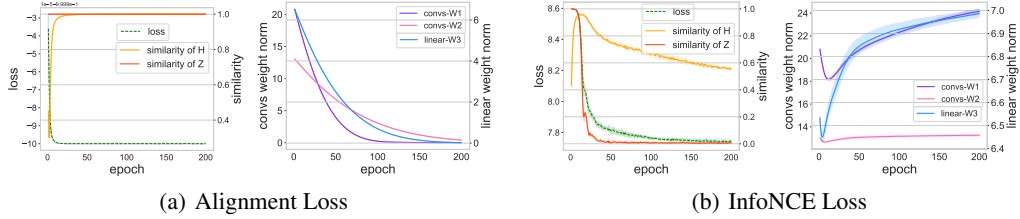

(a) Alignment Loss                    (b) InfoNCE Loss

Figure 3: Tendency of loss, average similarities of node representations $\mathbf{H}$ and $\mathbf{Z}$, and $L_2$ norms of weight matrices. We choose weight matrices of the first and the second convolutional layer (Convs-W1 and Convs-W2), and the first linear layer of the projection head (Linear-W3). Experiments are conducted on Cora with GRACE.

## E    Why No-negative GCL Not Collapse in the Graph Classification

In Section 5, we observe different phenomena in the graph classification and node classification. Specifically, in the graph classification task, GCL methods achieve decent performance in the no-negative setting, while the representations collapse in the node classification task. From the architecture perspective, we find in the graph classification task, the representations learned by the projector tend to be identity, while the representations learned by the encoder escape from collapse. We suspect that learning a collapsed solution is relatively easier for the global graph representation, which can be achieved solely by the projection head.

Here, we provide some empirical insights into these conjectures. Instead of researching how to make representations not collapse in the node classification, we choose to explore *when no-negative GCL collapses in the graph classification*. A straightforward method is stacking more layers within the encoder. The well-known over-smoothing issue in GNNs states that when the layer number increases, the representations will become identical and lose expressiveness [25]. This is exactly