# OpenReview forum: "Architecture Matters: Uncovering Implicit Mechanisms in Graph Contrastive Learning"
_NeurIPS.cc/2023/Conference — NeurIPS 2023 poster_

### Official Review · Reviewer_sd4m · 2023-06-21

**Soundness:** 4 excellent
**Presentation:** 3 good
**Contribution:** 3 good
**Rating:** 7
**Confidence:** 4

**Summary:**

In the paper "Architecture matters: uncovering implicit mechanisms in graph contrastive learning", the authors study common graph-level and node-level graph contrastive learning (GCL) methods based on graph convolutional networks (GCNs). The authors compare GCL with contrastive learning (CL) in computer vision (e.g. SimCLR), and highlight some important distinctions. In particular, they show that GCL based on GCNs does not require positive samples; can work without negative samples (if a particular normalization is used); and does not require any graph augmentations beyond adding Gaussian noise to the node features.

I liked this paper very much as it highlights and convincingly resolves some misconceptions in the GCL literature. The experiments are clear and convincing. I only have some minor questions and clarifications.

**Strengths:**

* Clearly written paper with clear arguments
* Convincing ablation experiments
* Shows the intricate interplay between architecture (GCN) and loss function (CL) in GCL -- something that I think has been neglected in the GCL literature

**Weaknesses:**

* Theorem 4.1: "Suppose the positive pairs are drawn from the ..." -- here it's unclear if this is actually similar to what happens in GRACE/GCA/ProGCL or not. You never describe the augmentations that they use in detail. I think you should comment here on how realistic is the assumption of this theorem.

* Table 4: I noticed that all accuracies in the 1st row ("Contrast") are lower than in Table 2. So MLP works worse then GCN. It could be interesting to comment on it. If GCN approximates the alignment loss (as per Theorem 4.1), then it seems GCN is not necessary if InfoNCE is used, and MLP could perform as good. But in reality, it performs a bit worse. Do you know why?

* Section 5.2 after equation (7): I wish there was some intuition provided here about what ContraNorm does, because equation (7) is rather mysterious.

* Section 5.2 shows that ContraNorm can more or less replace the uniformity loss in GRACE. I have two questions about it. First, does this only happen for the GCN encoder, or also with the MLP encoder? If you had two more rows in Table 5, for "NO Neg" loss with "MLP" and with "MLP+CN" encoders, would they differ between each other?

* Second, can ContraNorm replace the uniformity loss also in SimCLR? Or is this somehow graph-specific?

* Section 6 shows that Gaussian noise is almost sufficient as data augmentation in GRACE. Does this only happen using GCN? What if you use MLP? Would Gaussian noise still work?

* Table 7: what is the difference between "NO Aug" in Table 7 and "No Pos" in Table 2?

* Heterophily benchmarks in the Appendix use the Chameleon and Squirrel datasets but they are very problematic, see https://openreview.net/forum?id=tJbbQfw-5wv. Given the results of that paper, I am not sure if those two datasets should be used at all.

* You show that node-level contrastive learning like GRACE does not need positive pairs; does not need negative pairs either if ContraNorm is added; does not need topological graph augmentations (and augmentations anyway are not needed if no positive pairs are used). Can all of this be combined together? Can GRACE with GCN and ContraNorm be trained without positive AND without negative pairs? Does it even make sense?


Minor issues

* line 37: "performs well without any special designs in the no-negative setting" -- only when ContraNorm is used; this is not "without any special designs". I would reformulate.

* Section 3: The way you define G and tau(G), adding Gaussian noise to X does not qualify as data augmentation because X is not part of G. Maybe you want to define G as (V,E,X) ?

* Section 3: I would suggest to give some idea here of typically used graph augmentations.

* Section 3: I would suggest to comment in this section on the fact that f() is typically a GCN.

* Table 1: "No training" gives you 27% which is >> 10%. Is this a randomly initialized ResNet?

* "heterophily" is spelled "heteophily" in several places in the main text and in the appendix.

* For the experiments in Tables 2/3 and elsewhere, did you modify the source code of each method to switch the positive and negative pairs on and off? Or how was it implemented?

* line 231: "downstream results remain unaffected" -- this is only true for graph-level tasks (not for node-level), so I would clarify it in this sentence.

* line 249: this paper may be relevant here https://openreview.net/forum?id=ZgXfXSz51n

* line 254: maybe instead of "how does GCL manage to work" you should write "how can GCL be made to work" or something like that? It only works without negative samples for node classification if you use ContraNorm.

* line 308: I wish those augmentations were briefly defined here.

**Questions:**

See above

**Limitations:**

Adequately addressed

---

> ### Author Rebuttal · Authors · 2023-08-09
>
> We thank Reviewer sd4m for careful reading and appreciating the novelty and potential impacts of our work. Below, we summarize and address your main concerns.
>
> ---
>
> **Q1.** Comment on how realistic is the assumption of Thm 4.1. It is unclear whether it is similar to augmentations in GRACE/GCA/ProGCL or not.
>
> **A1.** The augmentations in GCN's implicit alignment loss (using neighbor nodes as positive pairs) differ from those in GCL methods (like node dropping and edge perturbation). However, Table C (Rebuttal PDF) demonstrates their **comparable performances**, suggesting the complementarity. Therefore, GCN's implicit alignment can replace positive samples under the No Pos setting, achieving good performance. This phenomenon could arise from shared domain priors: **neighboring nodes have similar labels, making slight perturbations of edges/nodes inconsequential for their class membership**.
>
> ---
>
> **Q2.** If GCN approximates the alignment loss, MLP could perform as well as GCN when using InfoNCE. But why does MLP perform a bit worse (Table 2&4)?
>
> **A2.** To compare the two methods, we assess 1) GCN + No Pos and 2) MLP + Contrastive loss with neighbor-induced positive samples (Thm 4.1). In Table D (Rebuttal PDF), **their performance is quite comparable** (80.82% vs. 83.43%), affirming our theoretical equivalence between graph convolution and neighbor-induced alignment loss. The marginal difference could arise because in GCN, there are some **nonlinearities interleaved with graph convolution**, enhancing its feature representation compared to pure MLP networks.
>
> ---
>
> **Q3.** Intuitions about ContraNorm, with Eq7 being rather mysterious.
>
> **A3.** We will add more intuitive explanations in the paper. Remind that as a gradient descent step of the uniformity loss [11], ContraNorm is written as:
>
> $$
> {\rm{ContraNorm(\mathbf{H})}} = \mathbf{H} - \alpha \mathbf{D}\mathbf{\tilde{A}}\mathbf{H}, \quad {\text{where }} \mathbf{\tilde{A}} = {\rm{softmax(\mathbf{D}\mathbf{H}\mathbf{H}^\top)}}, \mathbf{D}={\rm{diag}}(\mathbf{A}).
> $$
>
> Here, unlike the true adjacency matrix $\mathbf{A}$ given as a priori, $\mathbf{\tilde{A}}$ acts like a “fake” adjacency matrix computed online with the features $\mathbf{H}$, element-wisely, $\mathbf{\tilde{A}}_{ij}=\exp(d_i h_i^\top h_j) / \sum_k\exp(d_ih_i^\top h_k)$. Then, ContraNorm performs a ***reverse-direction* message passing** that substitutes the weighted neighbor information from the original features. Therefore, more similar nodes (larger edge weights in $\mathbf{\tilde{A}}$) get pushed further, explaining how it avoids feature collapse.
>
> ---
>
> **Q4.** ContraNorm more or less replaces the uniformity loss in GRACE. Does this only happen for GCN, or also for MLP?
>
> **A4.** In Table E (Rebuttal PDF), we show that CN could also boost MLP performance significantly (59.49%→73.19%) under the “No Neg” loss, and attain similar performance to MLP trained with contrastive loss (77.07%), which also aligns well with our theory and empirical observations on GCN. We will add these results in revision.
>
> ---
>
> **Q5.** Can ContraNorm replace the uniformity loss also in SimCLR? Or is this somehow graph-specific?
>
> **A5.** We conduct experiments replacing uniformity loss with ContraNorm and show the results in Table F (Rebuttal PDF). We find that ContraNorm can not replace the uniformity loss in SimCLR. It is a graph-specific technique.
>
> ---
>
> **Q6.** Would Gaussian noise still work when using MLP?
>
> **A6.** This is a very insightful question. We compare augmentations with MLP backbone. In Table G (Rebuttal PDF),  **while GCN exhibits similar performance with different augmentations, FM+EP notably surpasses Gaussian noise for MLP.** This also correlates with GCN's implicit alignment mechanism (Thm 4.1). Additional augmentations minimally affect GCN due to the existing alignment mechanism. However, for MLP without this implicit bias, graph-specific augmentations like FM+EP remain informative for learning proper graph invariance.
>
> ---
>
> **Q7.** Table 7: what is the difference between NO Aug in Table 7 and No Pos in Table 2?
>
> **A7.** *NO Aug* means applying no augmentations on graphs when generating views. *NO Pos* means the objective function is the uniformity loss. When applying no augmentations, the numerator of InfoNCE loss (Eq 1) is a constant with a fixed hyper-parameter $t$. However, **the augmentation procedure generates not only the positive samples but also the negative samples**, which explains the slight performance differences between *NO Pos* and *NO Aug.*
>
> ---
>
> **Q8.** Heterophily benchmarks Chameleon and Squirrel are problematic.
>
> **A8.** As suggested, we run experiments on the new proposed heterophily benchmarks. Overall, we find that GCN barely works on these graphs since **No Training usually performs as well as trained models** (Table H in Rebuttal PDF). Therefore, it suggests that GCL does not even work on these heterophily graphs, let alone its No Pos and No Neg variants.
>
> ---
>
> **Q9.** Can GRACE with GCN and ContraNorm be trained without positive AND negative pairs?
>
> **A9.** Removing both positive and negative samples renders InfoNCE loss empty, actually corresponding to the “No Training” baseline which typically performs much worse. Results in Table I (Rebuttal PDF) further show the performance remains poor even when adding ContraNorm to GCN. This is because despite the implicit alignment and uniformity mechanisms, **without any training objective, the model parameters in GCN+CN cannot be properly trained to fit the dataset**.
>
> ---
>
> **Q10**. Minor Issues.
>
> **A10.** For minor issues, we can not answer them one by one due to space limitations. Thanks for your constructive advice on writing, and we will revise our paper according to your suggestions. For the implementation, we include our source code in [link](https://anonymous.4open.science/r/Code_NeurIPS4041-305F).
>
> ---
>
> Hope our answers could address your concerns. Please let us know if you have additional questions.

---

> > ### Comment · Reviewer_sd4m · 2023-08-12
> > **Thank you**
> >
> > I thank the authors for their responses and clarifications, and for addressing many critical comments by doing additional experiments.
> >
> > I stand by my score of 7 and recommend accepting this paper. I read the other reviews and the authors' responses. I disagree with the assessment of the most critical reviewer.

---

### Official Review · Reviewer_4ia5 · 2023-07-01

**Soundness:** 2 fair
**Presentation:** 2 fair
**Contribution:** 2 fair
**Rating:** 4
**Confidence:** 4

**Summary:**

The paper explores graph contrastive learning (GCL) methods and identifies distinct phenomena that differentiate them from traditional visual contrastive learning (VCL) methods. It highlights that positive samples are not mandatory in GCL, negative samples are not necessary for graph and node classification, and data augmentations have a reduced impact. The paper emphasizes understanding the unique architecture of graph learning when designing GCL methods, rather than directly applying VCL methods.

**Strengths:**

1. The problem of contrastive learning on graphs is of significant importance.

2. The paper presents interesting conclusions, despite the possibility of some being incorrect.

3. The introduction provides a detailed overview of related work.


**Weaknesses:**

1. The paper should consider the potential differences in experimental setups between visual contrastive learning (VCL) and graph contrastive learning (GCL), such as variations in the use of supervised information ratios after contrastive learning. It is important to ensure consistency in experimental setups when drawing conclusions, as the differences may affect the validity of certain statements (e.g., the necessity of positive samples in GCL).

2. The lack of available source code hinders reproducibility and verification of the paper's findings.

3. The Introduction should explicitly highlight the contributions made by the paper.

4. Relevant baseline methods for graph contrastive learning (GCL), such as BGRL [1] and Auto-GCL [2], should be included in the comparison to ensure comprehensiveness.

[1] Shantanu Thakoor, et al. Large-scale representation learning on graphs via bootstrapping. ICLR 2022.

[2] Yihang Yin, et al. AutoGCL: Automated Graph Contrastive Learning via Learnable View Generators. AAAI 2022.


**Questions:**

Please see my comments in 'Weakness'.

**Limitations:**

Please see my comments in 'Weakness'.

---

> ### Author Rebuttal · Authors · 2023-08-09
>
> We thank Reviewer 4ia5 for careful reading and constructive comments. We summarize and address your main concerns in the following.
>
> ---
>
> **Q1**. The consistency in experimental setups when comparing visual CL and graph CL.
>
> **A1.** Across our experiments, we follow the standard setting and data splits in each domain. Specifically:
>
> - For node classification of GCL, we use a linear classifier with Adam as the optimizer. We randomly split the dataset and the train-val-test ratio is 1:1:8. We fix the learning rate as 0.01 and train 5000 epochs for all datasets.
> - For graph classification of GCL, we use a linear classifier with the regularization parameter selected in the range [0.001, 0.01,0.1,1,10,100,1000]. For data splitting, we adopt a 10-fold cross-validation.
> - For image classification of VCL, we use a linear classifier with SGD as the optimizer. The learning rate is 0.2 and scheduled with cosine annealing. The training epoch is 100. We use the default data splitting of CIFAR10 (train: test=50k: 10k).
>
> To further resolve your concerns on the consistency of evaluation setups, **we unify the evaluation settings for GCL and VCL**: we all train a linear classifier with an Adam optimizer, and randomly split the dataset with the same train-test ratio as 9:1. The results are shown in the following table.
>
> As can be seen, **the observations are consistent with the findings in our paper.** GCL and VCL present apparent differences in the properties of no-positives, no-negatives and random Gaussian noise augmentations. The supervised information ratio shows little effect on the conclusions. For node classification task, evaluations are under data splits of 1:9 in the paper and 9:1 here, but the findings are kept unchanged.
>
> *Node classification accuracy (%) with GRACE as backbone*
>
> | Loss | Encoder | Cora | CiteSeer | PubMed | Photo | Computers | Avg. |
> | --- | --- | --- | --- | --- | --- | --- | --- |
> | Contrast | GCN | 86.27 | 75.14 | 86.02 | 92.10 | 83.01 | 84.51 |
> | NO Training | GCN | 76.31 | 68.23 | 83.22 | 69.20 | 56.95 | 70.78 |
> | NO Pos | GCN | 86.35 | 74.05 | 85.50 | 91.42 | 83.33 | 84.13 |
> | NO Neg | GCN | 29.74 | 21.62 | 40.30 | 26.12 | 38.10 | 31.18 |
> | NO Neg | GCN + CN | 88.49 | 77.12 | 85.16 | 94.67 | 85.61 | 86.21 |
>
> *Graph classification accuracy (%) with GraphCL as backbone*
>
> | Loss | MUTAG | PTC_MR | PROTEINS | IMDB-B | IMDB-M | REDDIT-B | Avg. |
> | --- | --- | --- | --- | --- | --- | --- | --- |
> | Contrast | 91.58 | 71.43 | 78.21 | 76.40 | 52.93 | 88.70 | 76.54 |
> | NO Training | 86.32 | 65.71 | 72.14 | 70.80 | 44.13 | 75.30 | 69.07 |
> | NO Pos | 92.63 | 73.71 | 77.32 | 76.60 | 52.00 | 86.90 | 76.53 |
> | NO Neg | 91.58 | 69.71 | 78.75 | 77.80 | 52.80 | 87.00 | 76.23 |
>
> *Node classification accuracy (%) with GRACE as backbone with different augmentations*
>
> | Augmentation | Cora | CiteSeer | PubMed | Photo | Computers | Avg. |
> | --- | --- | --- | --- | --- | --- | --- |
> | NO Aug | 85.76 | 74.77 | 85.65 | 90.14 | 82.38 | 83.74 |
> | FM+EP | 86.27 | 75.14 | 86.02 | 92.10 | 83.01 | 84.51 |
> | Gaussian | 86.35 | 74.35 | 86.09 | 90.72 | 83.17 | 84.14 |
>
> *Image classification accuracy (%) with SimCLR backbone*
> | Loss | Augmentation | CIFAR-10 |
> | --- | --- | --- |
> | NO Training |Default| 22.5 |
> | InfoNCE |Default | 82.4 |
> | Alignment | Default | 18.6 |
> | Uniformity | Default | 20.3 |
> | InfoNCE | Gaussian | 38.6 |
>
> ---
>
> **Q2.** The lack of available source code hinders reproducibility and verification of the paper's findings.
>
> **A2.** We include our source code in this [link](https://anonymous.4open.science/r/Code_NeurIPS4041-305F). All our listed results are reproducible from this implementation.
>
> ---
>
> **Q3.** The Introduction should explicitly highlight the contributions made by the paper.
>
> **A3. We summarize our main contributions below and will add them in the revision:**
>
> - We perform comprehensive evaluations of popular GCL methods across various benchmarks, and find intriguing and general properties of GCL compared with VCL : 1) GCL works without positive samples; 2) GCL works without negative samples on graph classification task; 3) GCL shows less dependence on delicately designed augmentations, where random gaussian noise even works.
> - We reveal the reasons behind the intriguing properties of GCL by theoretically uncovering the interplay between contrastive learning objectives and model architectures: 1) We shed light on the implicit regularization mechanism of graph convolution by establishing its connection with a neighbor-induced alignment objective; 2) We show the collapse-preventing effect of ContraNorm in the no-negative setting by theoretically characterizing its connection with a neighbor-induced uniformity loss.
> - Our findings appeal to a re-examination of the real effectiveness of each component in GCL, and provide new insights for designing graph-specific self-supervised learning.
>
> ---
>
> **Q4**. Relevant baseline methods for graph contrastive learning (GCL), such as BGRL and Auto-GCL, should be included in the comparison to ensure comprehensiveness.
>
> **A4.** As suggested, we conduct experiments on Auto-GCL and BGRL for comparison. For Auto-GCL, we use a linear classifier instead of the SVM used in the paper for consistency. For BGRL, **it uses a non-contrastive loss without explicit usage of negative samples, so we can not conduct experiments under no-positive or no-negative settings.** Here we show the results under the original framework of BGRL.
>
> The results are shown in Table A and Table B (Rebuttal PDF). **For Auto-GCL, the observations on graph classification task are consistent with other methods (e.g. GraphCL) in our paper. For BGRL,  it achieves similar performance with methods using contrastive loss (e.g. GRACE, GCA).**  Thanks for your advice, and we will add Auto-GCL and BGRL as baselines in the revised version of our paper.
>
> ---
>
> Thanks for your comments and hope our answers could address your concerns. Please let us know if you have additional questions.

---

> > ### Comment · Reviewer_4ia5 · 2023-08-17
> > **To Authors**
> >
> > Thank you for your response. However, I share the concern raised by Reviewer nRNc, which pertains to the potential issue of over-claiming within this work. It's important to note that graphs and images possess fundamentally different data structures, and drawing conclusions about their distinctions might not be entirely credible. While I understand that the authors attempted to establish a rational basis using the 9:1 data split ratio, the appropriateness of this ratio for assessing both graph and image data remains questionable. Furthermore, it could be argued whether the experimental setups across the paper should be standardized. Given these concerns, I maintain my original score.

---

> > > ### Author Response · Authors · 2023-08-17
> > >
> > > Thanks for your response, and we address your further concerns below.
> > >
> > > ---
> > >
> > > **A1.** The potential issue of over-claiming within this work.
> > >
> > > **Q1.** In line with our response to Reviewer nRNc, we acknowledge that there are previous works that also discuss inductive bias, domain-specific augmentations and module designs. It was never our intention to assert being the first to find the importance of architectural inductive bias to contrastive learning, or we are the first to indicate no negatives are needed. Nevertheless, we want to emphasize that even after fully reviewing these prior works, our finding on 1) the non-necessity of positive samples and 2) the non-necessity of negative samples without any specific designs for graph classification is still never covered in prior works. Moreover, our theoretical explanation about the complementary role between alignment loss and graph convolution offers a fresh perspective in this domain. **We will add more explicit clarification on the contributions of our work in the revision and in particular, clarify the relationship between previous works and ours**. For more details please refer to our latest response to Reviewer nRNc.
> > >
> > > ---
> > >
> > > **A2.** It's important to note that graphs and images possess fundamentally different data structures, and drawing conclusions about their distinctions might not be entirely credible. While I understand that the authors attempted to establish a rational basis using the 9:1 data split ratio, the appropriateness of this ratio for assessing both graph and image data remains questionable.
> > >
> > > **Q2.** Regarding your concern about the distinction between CV and graph domains, we agree that it is hard to do a direct cross-domain comparison. Actually, we also did not do this kind of comparison. Instead, **we compared each method within its corresponding domain (NOT cross-domain comparison).** So, we can indeed conclude the following points from our experiments:
> > >
> > > - On the image classification task (Table 1), NoPos and NoNeg cannot outperform the random ResNet and is much worse than SimCLR. **All the experiments and comparisons are in the CV domain**.
> > > - On the graph classification task (Table 3), NoPos and NoNeg can statistically significantly outperform the random GNN and is comparable to GraphCL, etc. **All the experiments and comparisons are in the graph domain**.
> > > - On the node classification task (Table 2), NoPos and NoNeg+ContraNorm can statistically significantly outperform the random GNN and is comparable to GRACE, etc. **All the experiments and comparisons are in the graph domain**.
> > >
> > > These results showcase the distinct behaviors in the two domains when benchmarked against their domain-specific baselines. The logic here is that contrastive learning, which encompasses both positive and negative samples, behaves differently on images versus graphs. When positive samples are removed from image-based contrastive learning, the performance suffers significantly. In contrast, for graph-based learning, the removal of positive samples seems to have a negligible impact on performance. **It's important to clarify that we did not directly compare the performance of image-based contrastive learning with that of graph-based**. Instead, we took inspiration from these distinct behaviors to delve deeper into the unique characteristics of graph contrastive learning. All analytical experiments presented in our paper are conducted exclusively on graph data. We will state this clearer in the revision.
> > >
> > > ---

---

> > > ### Author Response · Authors · 2023-08-17
> > >
> > > **Q3.** Furthermore, it could be argued whether the experimental setups across the paper should be standardized.
> > >
> > > **A3.** As previously mentioned in A2: the majority of our experiments focus on graphs, with only the results in Table 1 and Table 6 pertaining to images. These tables serve primarily as a motivation to delve deeper into specific properties of graph contrastive learning.
> > >
> > > **For all our graph-related experiments, regardless of the task, we consistently adhere to standardized settings and data splits for GCL methods.** To detail:
> > >
> > > - For the node classification task, we follow the linear evaluation scheme in DGI [1], which is a common setting and also adopted by the original work of GRACE [2], GCA [3], and ProGCL [4].
> > > - For the graph classification task, we adopt the same evaluation procedure of InfoGraph [5], which is also commonly followed by the original work of GraphCL [6] and JOAO [7].
> > >
> > >
> > > **Reference**:
> > >
> > > [1]. Velickovic et al. Deep graph infomax. In *ICLR*, 2019.
> > >
> > > [2]. Zhu et al. Deep graph contrastive representation learning. *arXiv preprint arXiv:2006.04131*, 2020.
> > >
> > > [3]. Zhu et al. Graph contrastive learning with adaptive augmentation. In *WWW*, 2021.
> > >
> > > [4]. Xia et al. Progcl: Rethinking hard negative mining in graph contrastive learning. In *ICML*, 2022.
> > >
> > > [5]. Sun et al. Infograph: Unsupervised and semi-supervised graph-level representation learning via mutual information maximization. In *ICLR*, 2020.
> > >
> > > [6]. You et al. Graph contrastive learning with augmentations. In *NeurIPS*, 2020.
> > >
> > > [7]. You et al. Graph contrastive learning automated. In *ICML*, 2021.
> > >
> > > ---
> > >
> > > Hope the clarification above could address your concerns. Please let us know if there is more to clarify.

---

> > > ### Author Response · Authors · 2023-08-19
> > > **Your further inputs are appreciated!**
> > >
> > > Dear Reviewer 4ia5,
> > >
> > > We are truly appreciative of the feedback from you. Previously you mentioned a shared concern with Reviewer nRNc regarding potential over-claiming in our work. We addressed this issue in our response, and we are grateful to see that Reviewer nRNc has increased his rating after reading our clarifications. For other concerns raised by you, we also prepared a detailed response sent to you two days ago to address them. We were hoping to hear your feedback on them.
> > >
> > > We understand that everyone has a tight schedule, but we kindly wanted to send a gentle reminder to ensure that our response sufficiently addressed your concerns or if there are further aspects we need to clarify.
> > >
> > > Thank you for your time and consideration. We look forward to hearing from you soon.
> > >
> > > Best,
> > > Authors

---

### Official Review · Reviewer_RuBb · 2023-07-04

**Soundness:** 4 excellent
**Presentation:** 3 good
**Contribution:** 4 excellent
**Rating:** 7
**Confidence:** 4

**Summary:**

This manuscript demonstrates the correlation between the graph neural networks and the contrastive learning loss. Specifically, the authors illustrate that 1) because of the fusion of neighbor nodes caused by GNNs, neighbor nodes act as positive sample pairs (alignment); 2) incorporating an over-smoothing solution to enhance the representation diversity (and avoid feature collapses) can replace the negative pair loss (uniformity), while graph-level tasks are inherent immune to the feature collapse problem. The overall introduction is equipped with theoretical and experimental validations.

**Strengths:**

- The ideas are novel and worth sharing.
- The manuscript is well-written.
- The experiments are sufficient and promising.

**Weaknesses:**

W1: Line 86: The comparison with SCE is unclear. More explanations like the backbone comparison and learning objectives are expected.

W2: Line 156: The definition of NO Training is confusing. It is not clear what is no training here. Which self-supervised learning pipeline is used? Are the CL and downstream task co-trained or trained separately?

W3: The intuitive introduction of the reason why GNNs are inherent equipped with abilities of learning good representations is lacking in the section 1, causing evitable confusions. Lines 42 - 51 are unclear. To be specific, the intuitive explanations are expected for "the connection between graph convolution and a neighbor-induced alignment objective" and "incorporating a special normalization layer (what kinds of normalizations?) in the encoder can prevent (intuitively, how?) such collapse (intuitively, what is the collapse?)".

**Questions:**

Q1: Line 179: What is the correlation between the observed phenomena and that described in [1]?

Q2: Emphasizing the connection between over-smoothing and the feature collapse is expected.

[1] Yang, C., Wu, Q., Wang, J., & Yan, J. (2022). Graph neural networks are inherently good generalizers: Insights by bridging gnns and mlps. arXiv preprint arXiv:2212.09034.

**Limitations:**

Limitations and broader impacts are expected to be discussed separately.

Fail to mention licenses of datasets used in the paper.

---

> ### Author Rebuttal · Authors · 2023-08-09
>
> We thank Reviewer RuBb for your detailed reading and encouraging comments of our work on its novelty and insights. Below, we summarize and address your main concerns.
>
> ---
>
> **Q1.** Line 86: The comparison with SCE is unclear. More explanations like the backbone comparison and learning objectives are expected.
>
> **A1.** Thanks for your suggestion. Here we give a detailed comparison between SCE and ours. Generally speaking, SCE is a specific node-level GCL method, while we provide a comprehensive comparison for representative GCL methods on a range of datasets. Specifically, the two differ in:
>
> - **Tasks**. SCE only considers node classification tasks while we consider both graph and node classification tasks on various datasets.
> - **Backbone networks.** The backbone of SCE is a special multi-scale GCN variant, which adopts linear graph convolution (like SGC [1]) and aggregates multi-scale features at last (like JK-Net). In comparison, we adopt GCN for node classification tasks and GIN for graph classification tasks following the common practice.
> - **Learning objectives**. For training, SCE designs a new formulation of uniformity loss (the inverse of total pairwise distance) ($L_{unsup}$) and an L2 regularization on model weights ($L_2$):
> $$\mathcal{L}=\alpha \mathcal{L}_{\text {unsup }}+\beta \mathcal{L}_2$$
>
> $$\qquad = \frac{\alpha}{ \sum_{(v_i, v_j) \in \mathcal{N}} \Vert z_i-z_j \Vert^2}+\beta \Vert \theta \Vert^2.$$
>
> Therefore, SCE's unique backbone and objectives raise questions about the general applicability of their findings. Instead, theoretically and empirically, we demonstrate the general validity of the non-necessity of positive or negative samples across various tasks, backbones, and GCL objectives. We will add this comparison in the revision.
>
> ---
>
> **Q2.** Line 156: The definition of NO Training is confusing.
>
> **A2.** NO Training: **Randomly initialized backbone encoder with no pre-training applied.** We fix the encoder parameters and train a linear classification head on top. Further clarification will be provided in the revision.
>
> ---
>
> **Q3.** The intuitive explanations are expected for "the connection between graph convolution and a neighbor-induced alignment objective" and "incorporating a special normalization layer (what kinds of normalizations?) in the encoder can prevent (how?) such collapse (what is the collapse?)".
>
> **A3.** Following your suggestion, **we add concrete descriptions to the two statements you mentioned**:
>
> - We rigorously show that as graph convolution encourages neighbor nodes to have similar features during propagation, it implicitly minimizes a neighbor-induced alignment loss, which reveals an intriguing complementary relationship between the backbone network and the learning objective of GCL methods.
> - By incorporating a kind of normalization layer with the ability to drive nearby node features apart, we show that, without using the uniformity loss, a GCN encoder alone can prevent features from collapsing to a single point.
>
> ---
>
> **Q4.** Line 179: What is the correlation between the observed phenomena and that described in [1]?
>
> **A4.** This is a very intriguing question. In fact, our findings on the connection between graph convolution and the alignment objective can provide a natural explanation for why [1] works.  Specifically, [1] applies graph convolution to a trained MLP and observes improved performance. **This strategy, from our perspective, is amount to further training the MLP features with a neighbor-induced alignment loss (Thm 4.1) — only for a few steps (thus no severe feature collapse as “No Neg”) — which can help improve MLP’s performance.** We will add this discussion in the revision.
>
> ---
>
> **Q5.** Emphasizing the connection between over-smoothing and feature collapse is expected.
>
> **A5.** The over-smoothing refers to the phenomenon that when repeatedly applying the graph convolution, node features become indistinguishable. The feature collapse is observed in contrastive learning with the alignment loss alone, where all sample features collapse to a single point. **The formal equivalence established between graph convolution and the alignment loss (Thm 4.1) reveals that the two phenomena inherently describe the same thing**. We will emphasize this relationship in the revision.
>
> ---
>
> **Q6.** Limitations and broader impacts are expected to be discussed separately. Fail to mention licenses of datasets used in the paper.
>
> **A6.** Thanks for your constructive advice. Here we state the limitations and broader impacts separately and will add them in the revision.
>
> **Limitations.** Since the main goal of this work is to examine the roles of each component of GCL objectives, one limitation is that it does not propose a new GCL method. Nevertheless, we believe that the new findings in this work would be valuable for future GCL designs. Also, the paper does not examine other SSL paradigms on graphs, like masked modeling, which would be an interesting direction to explore.
>
> **Broader impacts.** As an understanding work of GCL learning mechanisms, this work does not include new models and potentially harmful findings. We think there will be little broader impacts.
>
> **Data License.** Below, we briefly present the licenses of used datasets and will add them in the revision.
>
> - **CIFAR10**: MIT License.
> - ****Cora, CiteSeer, PubMed****: These datasets are free to use for research and teaching purposes. See [LINQS](https://web.archive.org/web/20151007064508/http://linqs.cs.umd.edu/projects/projects/lbc/).
> - **Photo, Computers**: CC BY 4.0.
> - **Chameleon, Squirrel**: GNU General Public License v3.0.
> - **OGB-arxiv**: MIT License.
> - **MUTAG, PROTEINS, PRC-MR, IMDB-BINARY, IMDB-MULTI, REDDIT-BINARY**: Unknown. They are included in [TUDataset](https://chrsmrrs.github.io/datasets/).
>
> ---
>
> Thanks for your comments and hope our answers could address your concerns. Please let us know if you have additional questions.

---

> > ### Comment · Reviewer_RuBb · 2023-08-13
> >
> > Thank you for your insightful replies. Please revise the paper accordingly.
> >
> > Score update: $6\rightarrow 7$.

---

### Official Review · Reviewer_nRNc · 2023-07-26

**Soundness:** 3 good
**Presentation:** 3 good
**Contribution:** 2 fair
**Rating:** 5
**Confidence:** 5

**Summary:**

The current paper focuses on understanding the implication of design choices borrowed from visual contrastive learning (VCL) for graph contrastive learning (GCL). The authors argue that the need of positive samples is relatively weak in GCL; the architecture itself can implicitly act as a positive sample generator; negative samples are often unneeded as well, especially if apt architecture design choices are made; and augmentations can be very generic (e.g., gaussian noise in the node / edge features).


Rebuttals acknowledgment: I had apprehensions about the paper in regards to discussion of very related work and the lack of recognition of those papers. The authors engaged in a lively discussion that, if included in the paper, will address my concerns. I still believe the contributions are not very novel, but arguably worth sharing and repeating to the community. Accordingly, I raised my score to a borderline accept.

**Strengths:**

The re-evaluation of design choices in GCL is definitely a valuable target, given the extremely large number of frameworks in this field. Further, since most methods perform similarly well, I find the claims of this work valuable (though not necessarily novel).

**Weaknesses:**

My biggest apprehension with this paper is lack of novelty.

- Several prior works have shown the surprisingly good performance of random models and how performing GCL often leads to minimal improvements (depending on the task / dataset). I especially highlight the work by Trivedi et al. [1], who arguably were the first to thoroughly evaluate this claim.
- The claim on how generic augmentations work very well is again thoroughly investigated by Trivedi et al.[1], who show that the answer is much more subtle and depends on the application. In a fairly novel experiment, the authors create super-pixel MNIST graphs and perform GCL on such graphs. They show generic augmentations are significantly worse than domain-specific augmentations that account for the data-generating process. The authors further formalize this claim in a follow up work [2].
- I also note that the implicit benefits of architecture are a function of the fact that a significant portion of all self-supervised learning focuses on learning the right invariances that are useful for the data-generating process (see Von Kugelgen et al. [3]). As shown by Saunshi et al. [4], when the relevant invariance is directly enforced on the architecture, we can avoid the need of positive samples.

[1] Trivedi et al. (https://dl.acm.org/doi/abs/10.1145/3485447.3512200)

[2] Trivedi et al. (https://arxiv.org/abs/2208.02810)

[3] Von Kugelgen et al. (https://arxiv.org/abs/2106.04619)

[4] Saunshi et al. (https://proceedings.mlr.press/v162/saunshi22a.html)

**Questions:**

I would encourage the authors to clarify their work's novelty further.

**Limitations:**

See weaknesses.

---

> ### Author Rebuttal · Authors · 2023-08-09
>
> We thank Reviewer nRNc for carefully reading and appreciating the value of our work on the re-evaluation of designs in GCL. Below, we summarize and address your main concerns.
>
> ---
>
> **Q1**. Several prior works have shown the surprisingly good performance of random models and how performing GCL often leads to minimal improvements (depending on the task/dataset). I especially highlight the work by Trivedi et al. [1], who arguably were the first to thoroughly evaluate this claim.
>
> **A1**. Thanks for pointing out the relevant papers, but we are afraid that there are some misunderstandings about the main focus of this work. Indeed, Trivedi et al. [1] find random GNN models are also quite effective. Actually, the original GCN paper already found the amazing performance of random GNNs (Appendix A.1). However, we want to clarify that **this point is neither the focus nor the message of our paper**. Instead, **we show a contrary phenomenon that random models perform significantly worse than GCL** (e.g., 61.56% vs 71.29% with GraphCL), indicating the effectiveness of GCL training. Thus, our focus is **NOT to show the ineffectiveness of GCL (in our experiments, it really works), but to study the inner structure of the GCL objective, and the necessity of each component of GCL.** Remarkably, **we are the first to systematically show the non-necessity of each component of GCL, and none of these prior works [1-5] has found that GCL can work well without positive pairs or negative pairs.** We believe this finding and our explanations are valuable for understanding and advancing GCL methods.
>
> In short, we show that GCL training is helpful, but not all components are necessary; while Trivedi et al. argue for no GCL training at all. Therefore, the two works are sufficiently different. We will add this discussion in future updates. Please let us know if there is more to clarify.
>
> ---
>
> **Q2.** The claim on how generic augmentations work very well is again thoroughly investigated by Trivedi et al.[1], who show that the answer is much more subtle and depends on the application. In a fairly novel experiment, the authors create super-pixel MNIST graphs and perform GCL on such graphs. They show generic augmentations are significantly worse than domain-specific augmentations that account for the data-generating process. The authors further formalize this claim in a follow-up work [2].
>
> **A2.** First, we want to highlight **the key contribution of this work is the new findings on the non-necessity of the positive-pair loss and the negative-pair loss of GCL methods** (Sec 4 & 5). Further adding data augmentations (Sec 6) makes this discussion more complete: GCL depends less on complex augmentations because of the free-of-positives property. In comparison, Trivedi et al’s discussions [1,2] **only focus on data augmentations and do not study the individual roles of GCL objectives.**
>
> Second, **a subtle yet crucial difference lies in interpreting *domain-agnostic augmentations***. Trivedj et al. [1] regard *domain* as **types or sources of graphs, e.g., molecular graphs, web-document graphs, or super-pixel graphs.** However, in our paper, *domain* means **types of data, e.g., graph, image, or text.** Therefore, "*domain-agnostic augmentations*" actually implies different operations in Trivedi et al’s discussions [1,2] and our paper. For example, random edge editing is defined as domain-agnostic in [1], but viewed as domain-specific in our paper due to its graph-based nature.
>
> Third, **our research directions diverge regarding augmentations.** [1] focus on designing augmentation strategies with prior graph-type knowledge. In contrast, our study demonstrates that even simple augmentations won't **disrupt** the framework. Notably, random Gaussian noise causes a performance drop of nearly 7% (Computers in Table 7), **a discernible gap in the context of [1]**. But in our context,  **compared to VCL's steep 47% drop** (from 83.51% to 36.56% in Table 6), 7% is relatively marginal. Under such comparison, we think GCL is more tolerant of simple augmentations, and we attribute this robustness to graph convolution's alignment effect, an insight absent in [1].
>
> ---
>
> **Q3.** I also note that the implicit benefits of architecture are a function of the fact that a significant portion of all self-supervised learning focuses on learning the right invariances that are useful for the data-generating process (see Von Kugelgen et al. [3]). As shown by Saunshi et al. [4], when the relevant invariance is directly enforced on the architecture, we can avoid the need of positive samples.
>
> **A3.** Thanks for pointing out the related works [3,4]. Indeed, it is well-known that the CL objective is designed for learning invariance defined by data augmentations, e.g., [3]. On top of these known results, **the theoretical and empirical findings in our work further reveal the interplay and complementary effects between the CL objective and architectural designs on achieving this invariance (formally characterized in Theorem 4.1), which are new to this field**.
>
> Regarding the related work [4], after careful reading, we find that **Saunshi et al. [4]’s theory and experiments did NOT suggest the message that you mentioned**: “with invariance imposed on the architecture, we do not need positive samples”.  Specifically, in their experiments, **the perfect encoder f still requires training on the *standard CL objective that requires positive pairs* in order to perform well**. Remarkably, what their paper says is that “when the NN is misspecified (without inductive bias), CL may fail”, but NOT *“when NN has an inductive bias, we do not need positive pairs”*.
>
> Hope this clarification could ease your concerns. Please let us know if there is anything we overlooked here.
>
> ---
>
> Thanks for your comments and hope our answers could address your concerns. Please let us know if there is more to clarify. We are happy to address them during the discussion stage.

---

> > ### Comment · Reviewer_nRNc · 2023-08-13
> >
> > Thank you to the authors for their response. I spent quite a bit of time pondering over the answers and re-read the paper as well. I continue to remain apprehensive of the contributions and try to explain below why.
> >
> > 1. Relation to Trivedi et al. [1]: At a surface level, I agree with the authors that their evaluation covers ablating axes of GCL beyond Trivedi et al.'s. However, the fact that GCL or no augmentations GCL work better than random GCNs is a bit of an overstatement. Most benchmark datasets considered in this work are extremely simple (often binary classification) and small sized. Trivedi et al. claim that due to this, even though GCL does perform better than random models, the improvements only look large when represented in percentages and in actuality a bulk of the work is being done by the inductive bias of the model class. In the current paper, authors seem to state this result the opposite way, saying GCL models (both with and w/o augmentations) perform better than random models, but also attribute this property to the inductive bias of the architecture. I really do think both papers are getting at the same phenomenon broadly.
> >
> > 2. Inductive biases, role of projector, and normalization layers: Authors are correct that the paper I cited, Saunshi et al. [4], does not explicitly perform a no augmentation baseline---I misremembered, though I must add that the paper does have experiments where identity transforms, i.e., no augmentations, are used as positive samples in text domain experiments (see section 5.3). Nonetheless, the claim that effectiveness of contrastive learning is heavily influenced by inductive biases of the architecture, as heavily emphasized in the authors' paper, is what Saunshi et al. focus on. Their hash-augmentation experiments and related theoretical characterization are a perfect embodiment of this.
> >
> > 3. Negative samples and projector: I'm maybe moving goalposts with the following comment, but I'm slightly confused if this contribution is actually novel. Frameworks like SimSiam, BYOL, Barlow Twins etc. exist and show negative examples are not needed. Such frameworks involve use of projector / predictor and other tricks, however the graph SSL version of BYOL (https://arxiv.org/abs/2102.06514) already notes that a projector network is not needed in graph SSL. I'd say these results are sufficiently related to the current paper's claims.
> >
> > 4. Domain definition: This boils down to semantics in my opinion. I personally find the authors' claim that generic augmentations are sufficient in graph SSL unsatisfying---most SSL theory and empirical understanding papers strongly emphasize that augmentations relevant to the target task are needed to ensure useful representations are learned. We can't quite tell if what authors in this paper call "generic" is in fact relevant to the downstream task or not because the data-generating process (DGP) is unclear (I would again cite [3] here to emphasize that the DGP is really important to make comments on the relevance of an augmentation class). When prior works have tried truly generic augmentations in synthetic domains where the DGP is clear (e.g., [1, 3, 4]), they find generic augmentations are not useful in enabling good representations. So overall I appreciate the claims of generic augmentations in this paper, but I consider them (a) not quite well corroborated by the current experiments in the paper and (b) already well discussed in prior work.
> >
> > 5. I also add that authors' theoretical contribution on relating alignment loss with a pass of graph convolution is relatively straightforward---a convolution operation with appropriate assumption is a signal contracting operator (low pass filter). This is basically what the proof shows. I am stating this to argue that I do not consider the theoretical statements sufficient for accepting the paper---they're nice to have, but I'm not penalizing or valuing them for making a decision on the paper.
> >
> > Summary: I will continue to maintain my score for now.
> >
> > Edit: After internal discussions, I accept that the difference in task setup brings some novelty in terms of demonstrating generality of claims that in my opinion have been raised in prior work. I am increasing my score to 4 therefore.

---

> > > ### Author Response · Authors · 2023-08-15
> > >
> > > We thank Reviewer nRNc for your prompt reply and for reading our response carefully. We will address your further concerns.
> > >
> > > ---
> > >
> > > **Q1**. Relation to Trivedi et al. [1]: The fact that GCL or no augmentations GCL work better than random GCNs is a bit of an overstatement. Most benchmark datasets considered in this work are extremely simple (often binary classification) and small-sized.
> > >
> > > **A1.** We respectfully disagree with you on this point. After carefully reading the paper [1], we are afraid that **you might misread [1]’s messages, and as far as we see, their results are NOT contradictory to ours**. The key message of Sec 3.3 [1], is to show  ***“random GNNs are able to achieve non-trivial performance on several benchmark datasets”***, but **not “GCL cannot outperform random GNNs”** (as you thought). Actually, in Table 3 of [1], there are large-sized datasets on which GCL works better than random GNNs. For example, **NCI1 with 4110 graphs is a large-scale dataset** while GCL indeed works better than random GCNs (77.81% vs 70.65%). Further, a well-known GCL benchmark PyGCL [5] also shows that **GCL can achieve significant improvement over random GNNs on large-scale OGB graph classification benchmarks** (see Table I quoted below)**.** Similar results are observed on large-scale OGB node classification benchmarks in [6, 7], which are not binary classification (see Table II quoted below). Therefore, GCL is able to outperform random GNNs **in most cases** (not only the cases you pointed out like binary classification or small-sized). This is consistent with results in [1] and other well-known large-scale benchmarks in the literature [5,6,7].
> > >
> > > Table I. *Benchmark Metrics and Graph Classification / Regression Test Results (Cited from PyGCL [5])*
> > >
> > > | Dataset | ogbn-molhiv | PCQM4M-10K |
> > > | --- | --- | --- |
> > > | # Task  | Classification | Regression |
> > > | # Graphs | 41,127 | 10,000 |
> > > | Metric | ROC-AUC $\uparrow$ | MAE $\downarrow$ |
> > > | NO Aug | 55.86 +- 2.02 | 0.6041 +- 0.0464 |
> > > | Random Init (reproduced baseline) | 54.85 +- 4.13 | - |
> > > | InfoNCE loss | **65.18 +- 2.53** | **0.5287 +- 0.0117** |
> > >
> > > Table II. *Benchmark Metrics and Node Classification Test Results (Cited from BGRL [6], ProGCL [7])*
> > >
> > > | Dataset | ogbn-arxiv | PPI |
> > > | --- | --- | --- |
> > > | # Nodes  | 169,343 | 56,944 |
> > > | # Edges | 1,166,243 | 818,716 |
> > > | # Classes | 40 | 121 |
> > > | Metric | Accuracy $\uparrow$ | Accuracy $\uparrow$ |
> > > | Random Init | 69.90 +- 0.11 | 62.60 +- 0.20 |
> > > | GRACE | 72.61 +- 0.15 | **69.71 +- 0.17** |
> > > | ProGCL | **72.82 +- 0.08** | - |
> > >
> > > **References:**
> > >
> > > [5]. [Yanqiao Zhu, et, al. An Empirical Study of Graph Contrastive Learning. NeurIPS 2021, Track on Datasets and Benchmark.](https://datasets-benchmarks-proceedings.neurips.cc/paper/2021/file/0e01938fc48a2cfb5f2217fbfb00722d-Paper-round2.pdf)
> > >
> > > [6]. [Shantanu Thakoor, et, al. Large-Scale Representation Learning on Graphs via Bootstrapping. ICLR, 2022.](https://arxiv.org/pdf/2102.06514.pdf)
> > >
> > > [7]. [Jun Xia, et, al. ProGCL: Rethinking Hard Negative Mining in Graph Contrastive Learning. ICML, 2022.](https://arxiv.org/pdf/2110.02027.pdf)
> > >
> > > ---
> > >
> > > **Q2.** Relation to Saunshi et al. [4].
> > >
> > > - [4] does have experiments where identity transforms, i.e., no augmentations, are used as positive samples in text domain experiments (see section 5.3).
> > > - The claim that the effectiveness of contrastive learning is heavily influenced by inductive biases of the architecture, as heavily emphasized in the authors' paper, is what [4] focus on.
> > >
> > > **A2.** **We are afraid that you misunderstand [4]’s results again.** Here, the so-called “identity transform” actually refers to the augmentation strategy that randomly chooses a view among three views: the full text (identity), its left half, and its right half (see the first paragraph of Sec 5.3 [4]). We have to emphasize here that [4]’s “Split+Full” randomly returns one of the above views to form positive pairs, thus **there are actually no experiments with only identity augmentation (i.e., no augmentations) through the training** (the returned augmentation is totally **random!**)
> > >
> > > As for the comparison between [4] and ours on discussing the role of inductive bias, the key difference is that [4] only note the importance of inductive bias for **standard CL**, but they never find that inductive bias can replace the roles of positive pairs such that **NoPos GCL also works well**, which is the key new finding in this work. Notably, the inductive bias that we find is uniquely attributed to the **graph domain** with message-passing mechanisms, while [4] only discuss the **general inductive bias of CL in general CV & NLP domains**, which as we show, is clearly different from graph CL.
> > >
> > > ---

---

> > > ### Author Response · Authors · 2023-08-15
> > >
> > > **Q3.** Negative samples and projector: The graph SSL version of BYOL (BGRL) already notes that a projector network is not needed in graph SSL. These results are sufficiently related to the current paper's claims.
> > >
> > > **A3.** Indeed, similar to ours, BGRL also observes that GCL may require different model choices compared to VCL, like the projector. Nevertheless, **BGRL still inherits all the asymmetric designs in BYOL (predictor + EMA + stop-gradient).** In comparison, we are the first to find that **GCL methods can attain good performance on graph classification without these designs or negative samples**. This finding is a lot more surprising than BGRL’s free-of-projector finding. It is well known that without the projector, VCL can still achieve nontrivial performance (see SimCLR paper), but will totally collapse without asymmetry or go random without negative samples (see BYOL paper and ours). However, we show that in GCL, we can get good performance in graph classification even without any asymmetric designs or negative pairs.
> > >
> > > We believe that for people familiar with standard CL and its theory, this is mind-blowing and worth further exploration. This motivates us to share the findings with the community. Our paper shows that CL designs can vary so differently across domains, and we hope that it can lead us to rethink the roles of domain and model priors in CL.
> > >
> > > We will add these discussions on the relationship to BGRL’s findings in the revision.
> > >
> > > ---
> > >
> > > **Q4.** Domain definition: overall I appreciate the claims of generic augmentations in this paper, but I consider them (a) not quite well corroborated by the current experiments in the paper and (b) already well discussed in prior work.
> > >
> > > **A4.** We totally agree that when DGP is clear (i.e., with perfect domain knowledge), one can design perfect augmentations, which is definitely useful for the learned representations. While on real-world graphs, either DGP or perfect augmentation is not known. Under this background, the message we are trying to deliver in the paper is that “**simple augmentations can perform satisfying well compared to complex ones on real-world graphs, considering its simplicity**”, which is different from the prior work you mentioned.
> > >
> > > To further address your concerns, we additionally conduct experiments using a new kind of random noise, i.e., uniform noise. From the table below, we observe a similar performance on uniform noise to Gaussian noise. Considering these two different types of noise both hold on a diverse set of real-world graphs examined in our work, the observed phenomenon is convincing. Please note, **we are not claiming that it is a universal rule that holds for all graphs**, and **we have no intention to design new augmentations for graphs**. Actually, we are just finding some surprising phenomena that are worth noting and exploring. We will revise our statements to make them clearer in the revision.
> > >
> > > *Node classification test accuracy (%) on GRACE backbone with different augmentations*
> > >
> > > |  | Cora | CiteSeer | PubMed |
> > > | --- | --- | --- | --- |
> > > | NO Aug | 79.56 +- 2.18 | 71.83 +- 1.83 | 84.68 +- 0.58 |
> > > | EM + FP | 84.67 +- 1.39 | 73.47 +- 2.32 | 85.80 +- 0.16 |
> > > | Gaussian | 82.72 +- 2.38 | 72.60 +- 1.21 | 85.24 +- 0.61 |
> > > | Uniform | 81.25 +- 2.04 | 72.87 +- 2.07 | 85.39 +- 0.70 |
> > >
> > > ---
> > >
> > > **Q5.** The authors' theoretical contribution on relating alignment loss with a pass of graph convolution is relatively straightforward---a convolution operation with the appropriate assumption is a signal contracting operator (low pass filter). This is basically what the proof shows.
> > >
> > > **A5.** Despite that this reformulation only needs a few lines of proof, **it connects two seemingly irrelevant things (alignment objective in GCL and GNN propagation rules) that is never discussed before**, and it reveals what is the inductive bias of GNNs played in GCL, which could inspire some new designs in the future. Thus, we believe that its simplicity does not deny its novelty, as the review guide written by [Michael J. Black](http://ps.is.mpg.de/person/black). [8]: ***“Formulating a simple idea means stripping away the unnecessary to reveal the core of something. This is one of the most useful things that a scientist can do. A simple idea can be important.”.***
> > >
> > > Also, technically, there may be some misunderstandings of the proof. Here, the main technical challenge is to define a proper distribution of positive pairs such that it perfectly corresponds to the graph convolution. **It is not our goal, nor our tools, to show that graph convolution is a low-pass filter.** Actually, any graph filter (either high pass or low pass), can be reformulated into a corresponding alignment loss (with different definitions of positive pairs) in a similar fashion.
> > >
> > > [8]. [Novelty in Science - A Guide for Reviewers](https://perceiving-systems.blog/en/post/novelty-in-science).
> > >
> > > ---
> > >
> > > Thank you again for your reply. Please let us know if there is more to clarify.

---

> > > > ### Comment · Reviewer_nRNc · 2023-08-15
> > > >
> > > > Thank you for the detailed reply.
> > > >
> > > > I actively disagree with some claims the authors have made, e.g., that the proposed findings are surprising to people focused on science of self-supervised learning---I've published multiple papers on the topic and have kept up with literature quite properly, which is precisely why I am pushing back against what, to me, seems over-claiming in this work. I also note that some of my comments are being misrepresented; e.g., I never said Trivedi et al.'s results are contradictory to this work's! In fact, I said they already make similar claims and with similar experimental strategies, i.e., I deem the work similar, NOT contradictory! Finally, my comment on Saunshi et al.'s results simply stated that they do allow identity augmentations, not that identity is the only augmentation used in that work. I am unsure how I "again" misunderstand that paper's results.
> > > >
> > > > Regardless of the above, I can see the significant effort authors have put into engaging with what, in my opinion, were the most important related works for the theme of this paper and that do undercut or make the proposed results less surprising. I think if this discussion was properly included in the paper, it would significantly improve the quality of the work and bring me onboard with an accept. However, I do believe this will require a substantial rewrite and I am not comfortable recommending an accept without seeing an updated draft. I continue to maintain my score therefore.

---

> > > > > ### Author Response · Authors · 2023-08-16
> > > > >
> > > > > We greatly thank your prompt reply!
> > > > >
> > > > > ---
> > > > >
> > > > > **Q1.** I actively disagree with some claims the authors have made, e.g., the proposed findings are surprising to people focused on science of self-supervised learning---I've published multiple papers on the topic and have kept up with literature quite properly, which is precisely why I am pushing back against what, to me, seems over-claiming in this work.
> > > > >
> > > > > **A1.** First of all, we want to note that we are also very familiar with the “science of self-supervised learning”, having studied it for years and published multiple (5+) papers on top-tier venues about the general SSL theory. Thus, we are quite familiar with the literature, including those about invariance, DGP [3], and inductive bias [4]. So, our claim on the surprising finding in this work does not stem from unfamiliarity with the SSL theory literature. In fact, it is our previous experience on SSL theory that leads us to spot that GCL performs so differently on graph, and we are pretty sure that **none of the existing SSL theory work can rigorously explain its distinctive behaviors, esp. the NoPos and NoNeg experiments**. Our established connection between architectural inductive bias and CL objective could provide some insights into these mysteries, although much more work needs to be done to fully understand them. So, we believe that just like the surprise of BYOL that does not require any negative samples, **our findings of GCL (that no asymmetry is needed and no positive is needed) present new and intriguing challenges for the SSL theory community and are thus worth noting.**
> > > > >
> > > > > Also, although the inductive bias of GNN is well-known (mostly described as low-pass filters), **its complementary role to the alignment loss in contrastive learning is never revealed**. This relationship is crucial for understanding the unique behaviors of CL on graph data.
> > > > >
> > > > > ---
> > > > >
> > > > > **Q2.** Misrepresented my comments.
> > > > >
> > > > > **A2.**
> > > > >
> > > > > > I never said Trivedi et al.'s results are contradictory to this work! In fact, I said they already make similar claims and with similar experimental strategies, i.e., I deem the work similar, NOT contradictory!
> > > > > >
> > > > >
> > > > > There may be some misunderstandings here. In previous comments, you wrote, “***even though** GCL does perform better than random models, the improvements **only look large** when represented in percentages*”. From our understanding, you are saying GCL is actually not better than random models. When seeing the following sentence “*authors seem to state this result **the opposite way**, saying GCL models (both with and w/o augmentations) perform better than random models*”, the obvious word “opposite way” here further convinces us you are saying the two conclusions are contradictory.
> > > > >
> > > > > Suppose we skip over all the above misunderstandings on expression. Here, for your specific statement “*they already make similar claims and with similar experimental strategies*”, we respectfully have some dissenting opinions.
> > > > >
> > > > > - First, for the augmentation part, [1] identifies flawed practices in task-irrelevant graph augmentations and designs context-aware augmentations for synthetic graph datasets, while our work finds simple augmentations like random Gaussian noise also work on real-world graph datasets in GCL, totally different from VCL. **Briefly, [1] expects better task-relevant designs on graph augmentations, while we just find an intriguing phenomenon in augmentations of GCL with no intention to design new augmentations for graphs.**
> > > > > - Second, for the architecture inductive bias part, what [1] said is “*the inductive bias of GNNs can attain good results without training“*. In comparison, what we said is *“GCL works well without positive pairs because of the inductive bias of GNNs”*. Thus, **the two works reveal the benefits of inductive bias in two orthogonal directions: no training (no parameter update) and GCL training (there are parameter updates but without using positives)**. What is more important here, **we demonstrate what is the inductive bias of GNNs and how it works**, i.e., the inductive bias of GNNs is a kind of alignment during model training. So our finding on the inductive bias in GCL training is indeed new and valuable to this field.
> > > > >
> > > > > In conclusion, we think the main claims and experiments strategies of [1] and our work are not similar.
> > > > >
> > > > > > my comment on Saunshi et al.'s results simply stated that they do allow identity augmentations, not that identity is the only augmentation used in that work.
> > > > > >
> > > > >
> > > > > Maybe we misunderstand your comments to some extent. You wrote "*identity transforms are used as positive samples*". It is somewhat misleading. Precisely speaking, in [4], it might mean "identity transforms are used as **a part of** positive samples". However, in our work, we **remove** **all** positive samples. The two experiments are totally different. Therefore, **we don't think this experiment in [4] can serve as strong evidence for denying the novelty of our NO Pos experiments.**
> > > > >
> > > > > ---

---

> > > > > ### Author Response · Authors · 2023-08-16
> > > > >
> > > > > **Q3.** Update draft.
> > > > >
> > > > > **A3.** Unlike ICLR, NeurIPS does not allow the change of PDF during the rebuttal in accordance with NeurIPS guidelines. However, **we approached the rebuttal process with a sincere intent to address concerns, and we've made substantial efforts to prepare the rebuttal, for example, the differences from previous works.** These explanations have already been well discussed and can be seamlessly integrated into the main content of the paper upon revision.
> > > > >
> > > > > We truly value your feedback. We are confident that, given the opportunity for revision, we can easily address your concerns and enhance the overall quality of the paper. **In this context, we have briefly outlined the relationships between our work and references [1-4] below, with the intent to incorporate these discussions more extensively in our subsequent draft.**
> > > > >
> > > > > For the inductive bias of architecture, [4] proposes a general theoretical framework showing the importance of architecture inductive bias to standard contrastive learning. [1] presents comparable performances between GCL and untrained GCNs on some relatively simple benchmarks, showing the existence of inductive bias of GCL. In contrast, our work firstly uncovers what exactly the inductive bias of GNNs is by exploring the dynamic interplay between the GNN architecture and the contrastive optimization objective during training. For data augmentations, [1, 2] identify limitations in existing task-irrelevant graph augmentations, and expect better practices in augmentations considering the graph-domain knowledge. [3] also proves a good augmentation should keep the semantics of the data invariant. However, our works are not intended to design stronger data augmentations, but we go another way that finding simple augmentations like random Gaussian noise also work on real-world graph datasets in GCL, while it causes a steep performance drop in VCL.
> > > > >
> > > > > ---
> > > > >
> > > > > Thank you again for your reply. Please let us know if there is more to clarify.

---

> > > > > > ### Comment · Reviewer_nRNc · 2023-08-16
> > > > > >
> > > > > > I've read the latest comments--thank you for clarifying the misunderstandings that seem to be occurring here. I have two broad notes:
> > > > > >
> > > > > > 1. In my opinion, the primary contributions are valuable for the community to be aware of. Based on my familiarity with relevant work, however, the proposed results come off especially less surprising. I strongly encourage the authors to use language that is fairly claiming their addition to known knowledge of Graph SSL and not over-claim. For example, while I can see the point that no positives were needed for the task explored in this work has been exhaustively explored by the authors, focusing on this point is akin to removing attention from the fact that the writing suggests the authors are first to indicate, e.g., no negatives are needed. Similarly, while I'm neutral to authors including the relation between a GraphConv layer and implicit alignment, I strongly emphasize that the simplicity of the argument makes it more of a hypothesis than a full-fledged certainty that this relationship is the reason why GCL does not (at times) need positive samples. The writing is too assertive in my opinion around that section. The authors' comments above also come off as this result being "the reason", instead of a hypothesis. Finally, I would bring to authors' attention two more works that may be relevant to their results on removal of projection layer and use of a normalization layer in its stead (see below). I should point out that use of a normalization layer after a projector is fairly common when negatives are not present in the training pipeline (SimSiam's original implementation did that and it really reduces performance when this strategy is not applied on ImageNet).
> > > > > >
> > > > > > (i) http://openaccess.thecvf.com/content/ICCV2021/html/Hua_On_Feature_Decorrelation_in_Self-Supervised_Learning_ICCV_2021_paper.html
> > > > > >
> > > > > > (ii) https://arxiv.org/abs/2102.06810
> > > > > >
> > > > > > 2. I would emphasize the relationship with prior works and differences in what's new within this paper throughout the text, i.e., go beyond a short discussion in say related work section (I understand the para in last comment by the authors is only indicative of what they intend to include).
> > > > > >
> > > > > > As per paper decision, I understand the authors' point that the paper cannot be updated and share the frustration. I genuinely would have recommended accept if the edits could be made and I could review the paper. However, without that, I cannot in good faith recommend an accept. I leave it to the AC to make a call therefore. My conclusion on the paper is that the results are valuable, if not necessarily very novel, but the required text update to ensure the article is appropriate in its claims requires what to me seems a substantial rewrite.

---

> > > > > > > ### Author Response · Authors · 2023-08-17
> > > > > > >
> > > > > > > Thanks for your careful reading of our response and for the valuable time put into this engaging discussion.
> > > > > > >
> > > > > > > ---
> > > > > > >
> > > > > > > **Q1.** I strongly encourage the authors to use language that is fairly claiming their addition to known knowledge of Graph SSL and not over-claim.
> > > > > > >
> > > > > > > **A1.** Thanks for your suggestions. We finally get your meaning on the tone of our writing. The intent was never to overstate our contributions, and we acknowledge that the current phrasing might lead to unintended conclusions. We address them in the following points and will ensure a more careful presentation in the revision.
> > > > > > >
> > > > > > > > For example, while I can see that no positives are needed for graph tasks has been exhaustively explored by the authors, focusing on this point is akin to removing attention from the fact that the writing suggests the authors are first to indicate, e.g., no negatives are needed.
> > > > > > > >
> > > > > > >
> > > > > > > Regarding our No Pos experiment, we're grateful for your recognition of its novelty. For the No Neg part, we wish to clarify that our intent was not to claim we are the "first to indicate no negatives are needed". Rather, our primary assertion is the observation that "for graph classification, CL can even work without negative or any specific designs", especially given the prevailing belief in the SSL community regarding the necessity of asymmetry to prevent collapse (SimSiam and Tian et al (ii)).
> > > > > > >
> > > > > > > For example, we will revise our paper as below:
> > > > > > >
> > > > > > > - Line 46-48: “Second, we highlight the role of the projection head playing in negative-free GCL for the graph classification task, where we find the projection head implicitly selects a low-rank feature subspace to satisfy the loss.” →  “Second, we highlight the role of the projection head playing in **GCL without negative samples or any specific designs**, where we find for graph classification, the projection head implicitly selects a low-rank feature subspace to satisfy the loss.”
> > > > > > > - Line 203-205: ”In particular, we observe that negative samples are dispensable for the graph classification task, whereas in the node classification task, simply removing them may not be sufficient.” → “In particular, we observe that negative samples are dispensable **without any specific designs** for the graph classification task, whereas in the node classification task, simply removing them may not be sufficient.”
> > > > > > >
> > > > > > > > I strongly emphasize that the simplicity of arguments makes it more of a hypothesis than a full-fledged certainty that this relationship is the reason why GCL does not (at times) need positive samples. The writing is too assertive in my opinion around that section. The authors' comments above also come off as this result being "the reason", instead of a hypothesis.
> > > > > > > >
> > > > > > >
> > > > > > > Indeed, our theoretical explanations assume that GNNs only contain graph propagations and ignore nonlinear feature transformations. So this analysis is rigorous for linear GNNs (like SGC). When applied to nonlinear GNNs, it could be better framed as hypotheses. While linear GNNs, like SGC, often showcase comparable performance and are commonly adopted for theoretical analyses (e.g., in studies of over-smoothing [1]). Thus, our analysis on GCL with linear GCNs is also insightful for understanding GCL with nonlinear GCNs. We acknowledge the need to present our analyses more cautiously and will tune down the claims in the revision.
> > > > > > >
> > > > > > > For example, we will revise our paper as below:
> > > > > > >
> > > > > > > - Line 9-11: “By uncovering the implicit mechanism of the architecture to contrastive learning, we theoretically give explanations for the above intriguing properties of GCL.” → “By uncovering **how the implicit inductive bias of GNNs works** in contrastive learning, we theoretically **provide insights into** the above intriguing properties of GCL.”
> > > > > > >
> > > > > > > > Finally, I would bring to attention two more works that may be relevant to removal of projection layer and use of a normalization layer in its stead. I should point out that using a normalization layer after a projector is fairly common without negatives in the training (SimSiam's original implementation did that and it really reduces performance when this strategy is not applied on ImageNet).
> > > > > > > >
> > > > > > >
> > > > > > > First, we note the key difference to SimSiam is that SimSiam still needs asymmetry (predictor and stop gradient), and only the projector or normalization in SimSiam cannot avoid collapse; while we show that it is not the case in GCL. While we do observe similarities with ContraNorm and whitening in Hua et al., our unique observation lies in demonstrating that "GCL can work even without any normalization in graph classification task (in node classification task, normalization is still needed when no negatives)", a result remaining enigmatic currently. We understand the importance of delineating our findings in the context of these prior contributions and will provide a clearer distinction in the revision.
> > > > > > >
> > > > > > > Reference:
> > > > > > >
> > > > > > > [1] Li et al. Deeper insights into graph convolutional networks for semisupervised learning. AAAI. 2018.

---

> > > > > > > ### Author Response · Authors · 2023-08-17
> > > > > > >
> > > > > > > ---
> > > > > > >
> > > > > > > **Q2.** I would emphasize the relationship with prior works and differences in what's new within this paper throughout the text, i.e., go beyond a short discussion in say related work section (I understand the para in the last comment by the authors is only indicative of what they intend to include).
> > > > > > >
> > > > > > > **A2.** Indeed, a comprehensive discussion is essential to distinctly highlight the novel contributions of our paper. We will sort out the differences from these relevant works and mention them explicitly throughout the paper when stating the claims, not just in the related work section. **With the current sufficient and lively discussions, we believe that this revision is relatively easy and we are confident to deliver a fair discussion in the revision**.
> > > > > > >
> > > > > > > For example, besides discussions on related work, we will revise our paper as below:
> > > > > > >
> > > > > > > - Line 52-55: “These intriguing distinctive properties of GCL reveal that the design of contrastive learning can be very domain-specific. Importantly, the contrastive algorithm has an implicit interaction with the architecture.” **→** “These intriguing distinctive properties of GCL reveal that the design of contrastive learning can be very domain-specific. Importantly, **as also shown in previous works [1,4],** the contrastive algorithm has an implicit interaction with the architecture.”
> > > > > > > - Line 103-105: “Instead of designing elaborate augmentations for improving performance, in this paper we explore how robust GCL is on simple domain-agnostic augmentations like random Gaussian noise.” **→** “**There are also works identifying limitations in existing task-irrelevant graph augmentations, and expect better practices in augmentations considering the graph-domain knowledge [1,2,3].** However, our works are not intended to design stronger data augmentations, but we go another way that finding simple augmentations like random Gaussian noise also work on real-world graph datasets in GCL, while it causes a steep performance drop in VCL. ”
> > > > > > >
> > > > > > > ---
> > > > > > >
> > > > > > > Hope the clarification above could address your concerns. Thanks for your time again and we are happy to take your further questions.

---

> > > > > > > > ### Comment · Reviewer_nRNc · 2023-08-18
> > > > > > > >
> > > > > > > > I thank the authors for their sincere efforts and engagement. The indicative rewriting edits authors mention in the last comment are a step in the right direction, but arguably need to become fairer. For example, merely saying "there are also works identifying limitations in existing task-irrelevant graph augmentations...." barely scratches the surface of what those papers show. It is necessary to be precise on how their experimental setup is similar. Those papers do have real world graph datasets as well after all. I also add that the note "generic augmentations cause performance drop in VCL" is entirely a matter of perspective. Gaussian noise is in fact extremely important as an augmentation in all VCL frameworks out there, leading to performance drop if not included. Similarly, one could argue color jittering is a fairly generic augmentation. From this viewpoint, there will be some setting where Gaussian noise is actually a "domain-relevant" augmentation for graphs as well---precisely the point I was trying to highlight with notes on the data-generating process and the work of Von Kugelgen et al. Accordingly, I would rewrite the phrasings on your Gaussian noise result to something like: "maybe, the architecture is playing some implicit bias, or maybe the addition of Gaussian noise is actually the correct augmentation! We don't and won't know unless we have grasp over the data-generating process. However, given that Gaussian noise works, empirically it's a worthwhile augmentations for people to try in practice."
> > > > > > > >
> > > > > > > > Broadly, I'm happy with the effort authors have put in to reconsider the purview of their work. I'm accordingly increasing my score to 5. I emphasize that it is extremely important, if accepted, the writing is improved in the final version to take all comments above into account. I'm essentially improving my score recommendation on "good faith" and sincerely hope the appropriate edits are made.

---

### Author Rebuttal · Authors · 2023-08-10

Here we provide supplementary materials for rebuttal.

---

### Decision · Program_Chairs · 2023-09-21

**Decision:**

Accept (poster)

**Comment:**

The paper considers the design choices used in Graph Contrastive Learning (GCL) that have been borrowed from Visual Contrastive Learning (VCL) and how they differ across these domains. In particular, they rigorously quantify the role of positives and negatives samples, and augmentations in GCL through several experiments and theoretical analysis.

The insights are interesting and valuable to the community, as noted by all reviewers, and the authors have put in extensive effort to address the reviewers' concerns. Therefore, I recommend accepting the paper. However, given that several reviewers have raised over-selling as a concern, it is extremely important that in the final manuscript, the authors address the concerns of the reviewers that were raised in the reviews/discussions (as they have in their responses), discuss additional relevant prior work, and adjust the exposition to fairly represent their contributions.